# On the role of overparameterization in off-policy Temporal Difference learning with linear function approximation

**Valentin Thomas**
Mila, Université de Montréal
`vltn.thomas@gmail.com`

## Abstract

Much of the recent successes of deep learning can be attributed to scaling up the size of the networks to the point where they often are vastly overparameterized. Thus, understanding the role of overparameterization is of increasing importance. While predictive theories have been developed for supervised learning, little is known about the Reinforcement Learning case. In this work, we take a theoretical approach and study the role of overparameterization for off-policy Temporal Difference (TD) learning in the linear setting. We leverage tools from random matrix theory and random graph theory to obtain a characterization of the spectrum of the TD operator. We use this result to study the stability and optimization dynamics of TD learning as a function of the number of parameters.

## 1 Introduction

Occam's razor principle states that among many plausible explanations, the simplest one is the most likely to be true. In statistics and machine learning, this is often interpreted as models with a restricted number of parameters should be privileged as they generalize better (Akaike, 1974). This view is supported by the classical *bias-variance trade-off* which implies that models that are too flexible are bound to overfit (Geman et al., 1992). Yet, deep neural networks with more parameters than training data points achieved remarkable performance in many domains (LeCun et al., 1989; Krizhevsky et al., 2012; Devlin et al., 2018), challenging this conventional wisdom. A significant understanding of this apparent tension has been reached in recent years. A large body of work using tools from *random matrix theory* has shown that, in models with random features, overparameterization does not lead to overfitting but instead better generalization (Belkin et al., 2020). This phenomenon is known as *double descent* and unifies the traditional and modern understanding of bias and variance. Importantly, Kuzborskij et al. (2021) has shown that the double descent phenomenon is in large part explained by purely optimization rather than label noise. Following their insight, we mainly take an optimization approach in the rest of our paper.

While we now have a theory of overparameterization in supervised learning, the theoretical analysis of overparameterization in the reinforcement learning (RL) setting is largely unexplored (Xiao et al., 2021). Indeed, RL poses several new additional challenges. First, the agent has to interact with an *environment* whose structure may be complex. Second, each state may not be sampled with equal probability while each datapoint is usually sampled uniformly in supervised learning. In this work, we focus mainly on *policy evaluation*, i.e evaluating the expected return of a fixed policy. More precisely, we study the behavior of Temporal Difference learning (Sutton, 1988) which has become ubiquitous for policy evaluation in modern Deep RL (Mnih et al., 2013; Haarnoja et al., 2018).

Our contributions are, in the limit of large number of states and parameters: **1)** We introduce a class of random graphs for which we can compute analytically the spectrum of the TD operator

36th Conference on Neural Information Processing Systems (NeurIPS 2022).

when using uniform off-policy sampling, **2)** we derive approximations for the largest and smallest eigenvalue in the non-uniform case. Furthermore, we analyze the behavior of the traditional Mean Square Bellman Error (MSBE) when learning with TD(0) and show that **3)** it exhibits a double descent phenomenon similar to supervised learning but with a multiplicative factor depending on the discount factor in the overparameterized regime and **4)** the asymptotic residual MSBE displays a peaking behavior near the interpolation threshold (when the number of parameters approaches the number of states) that is specific to RL. Finally, **5)** we showcase how the non-stationarity of the policy can make the optimization of the value function unstable. All of these points are motivated theoretically and empirically validated on small gridworlds MDPs.

## 2 Background

### 2.1 Value estimation

**Definition:** We place ourselves in the Markov Decision Process (MDP) setting (Puterman, 2014) defined by $\{\mathcal{S}, \mathcal{A}, \mathbf{P}_{sa \to s}, \mathbf{r}, \gamma, p_0\}$, where $\mathcal{S}$ and $\mathcal{A}$ denote the finite state and action spaces, $\mathbf{P}_{sa \to s} \in \mathbb{R}^{|\mathcal{S}| \cdot |\mathcal{A}| \times |\mathcal{S}|}$ is the environment transition probability matrix and $\mathbf{r} \in \mathbb{R}^{|\mathcal{S}| \cdot |\mathcal{A}|}$ denotes the (deterministic) reward vector. $\gamma \in [0, 1[$ is the discount factor, and lastly, $p_0$ is the initial state distribution. For a given *policy* $\pi$, i.e a distribution over actions given a state, the goal of value estimation is to be able to estimate the expected discounted return $\mathbf{q}^\pi(s, a) = \mathbb{E}_{\mathbf{P}_{sa \to s}, \pi}\left[\sum_t \gamma^t \mathbf{r}(s_t, a_t) | s_0 = s, a_0 = a\right]$. This quantity is also known as the state-action value or the Q-function. Furthermore, it can be written as the solution of the *Bellman equation* (Bellman, 1954)[1]: $\mathbf{q}^\pi(s, a) = \mathbf{r}(s, a) + \gamma \mathbb{E}_{\mathbf{P}_{sa \to s}, \pi}[\mathbf{q}^\pi(s', a')]$. By defining $\mathbf{P}^\pi \in \mathbb{R}^{|\mathcal{S}| \cdot |\mathcal{A}| \times |\mathcal{S}| \cdot |\mathcal{A}|}$ the state-action to state-action transition matrix, we can write it in matrix form

$$\mathbf{q}^\pi = \mathbf{r} + \gamma \mathbf{P}^\pi \mathbf{q}^\pi \tag{1}$$

**Linear function approximation:** While Equation (1) can be solved efficiently when $\mathbf{r}, \mathbf{P}^\pi$ are known and when $n = |\mathcal{S}| \cdot |\mathcal{A}|$ is not too large, in many cases this cannot be computed. One solution is to introduce a parameter vector $\theta \in \mathbb{R}^p$ and a feature matrix $\Phi \in \mathbb{R}^{n \times p}$ and optimize $\theta$ to approximate $\mathbf{q}^\pi$ as $\mathbf{q}^\pi \approx \Phi\theta$. This is referred to as the Linear Function Approximation (LFA) setting.

**Temporal Difference with LFA:** Estimating the Q-function in the Linear Function Approximation setting can be done using the well-known algorithm TD(0) (Sutton, 1988) whose expected update is

$$\theta_{t+1} = \theta_t + \eta \Phi^\top \Xi \left(\mathbf{r} + \gamma \mathbf{P}^\pi \Phi \theta_t - \Phi \theta_t\right) \tag{2}$$

where $\Xi$ is the off-policy distribution matrix, a $n \times n$ diagonal matrix with positive entries summing to 1, weighting each state-action by their probability of being sampled. Contrary to gradient descent with an appropriate step-size, TD(0) is not guaranteed to converge as the matrix $\Phi^\top \Xi \left(\mathbf{I}_n - \gamma \mathbf{P}^\pi\right)\Phi$ might have a negative eigenvalue. This phenomenon is known as the *deadly triad* as it can only happen when we are (i) off-policy, (ii) using function approximation and (iii) bootstrapping on the next value as TD(0) does (Baird, 1995; Tsitsiklis & Van Roy, 1997; Van Hasselt et al., 2018).

### 2.2 Random Matrix Theory

**Wishart matrices:** Extremely relevant in the study of least squares with random features are matrices from the *Wishart distribution* (Wishart, 1928). These matrices can be written as $\frac{1}{n}\Phi^\top\Phi$ for $\Phi$ a $n \times p$ matrix with entries of zero mean, unit variance and bounded 4th moment sampled independently and identically distributed. In that case, when $p, n \to \infty$ and the ratio converges to a finite limit $\lim_{p,n \to \infty} p/n = \rho \in ]0, +\infty[$, the empirical spectral distribution converges to the Marchenko-Pastur (Marčenko & Pastur, 1967) (MP) distribution whose density can be written as

$$d\mu(x) = \max\{1 - 1/\rho, 0\} \, 1_{\{0\}}(x)dx + \frac{1}{2\pi\rho x}\sqrt{(x - \lambda_{\min}^+)(\lambda_{\max} - x)} \, 1_{[\lambda_{\min}^+, \lambda_{\max}]}(x)dx \tag{3}$$

where $1_A(\cdot)$ is the indicator function, valued at 1 if $x \in A$, 0 otherwise and $\max\{1 - 1/\rho, 0\} \, \delta_{\{0\}}(x)$ are the 0 eigenvalues, which exist when $p \geq n$, $\lambda_{\max} = (1 + \sqrt{\rho})^2$, the largest, and $\lambda_{\min}^+ = (1 - \sqrt{\rho})^2$, the smallest non-zero eigenvalue.

---

[1]Note that we study here the expectation Bellman equation, not the optimality Bellman equation which would feature a $\max$ operator over the next action.

**Wigner-type matrices:** Wigner (1955) historically introduced and studied symmetric $n \times n$ matrices of the form $\mathbf{W}$ where $\mathbf{W}_{ij}, i \geq j$ are sampled i.i.d from a law of mean 0 and unit variance for $i \geq j$[2]. $\frac{1}{\sqrt{n}}\mathbf{W}$ has a real spectrum which converges to the *semi-circle* distribution $d\mu(x) = \frac{1}{2\pi}\sqrt{4-x^2}\,1_{[-2,2]}dx$. When non-symmetric, the spectrum converges to the complex unit disk $\{z \in \mathbb{C}, |z| \leq 1\}$[3] (Tao et al., 2010).

## 2.3 Random graphs

Random graphs are useful models to infer structural or spectral properties of typical graphs. The earliest theoretical analyses of random graphs that we are aware of were done independently by Erdős et al. (1960) and Gilbert (1959). The model introduced in Erdős et al. (1960) is a random graph where each of the $n$ nodes is connected at random to $d$ others. The second model (Gilbert, 1959) is slightly different as each node is independently connected to any other with a probability $\mu \in ]0,1[$. We refer to the first one as the $d$-regular Erdős-Rényi model $\mathbb{G}(n,d)$ and the second one as the Erdős-Rényi-Gilbert $\mathbb{G}(n,\mu)$ model.

Given the adjacency matrix $\mathbf{A}$, with $\mathbf{A}_{ij} = 1$ if $i$ is connected to $j$, it is possible to construct a matrix $\mathbf{P}$ by renormalizing the rows of $\mathbf{A}$ so that $\mathbf{P}$ is a stochastic matrix (positive entries and rows summing to 1). $\mathbf{P}$ corresponds to the Markov transition matrix of a random walk on the graph and is connected to the *(left)-normalized Laplacian* $\mathbf{L}$ by $\mathbf{L} = \mathbf{I}_n - \mathbf{P}$. For many classes of random graphs, the spectrum of these quantities is known (Zhao, 2012; Tran et al., 2010).

# 3 Spectrum of TD: from random graphs to spiked models

## 3.1 From Ordinary Least Squares to TD(0)

Random matrix models have been used in supervised learning, especially in the simple least squares model (Cun et al., 1991; Hastie et al., 2019; Derezinski et al., 2020), to explain phenomena arising in deep learning. Indeed, while much simpler than large neural networks, these models can shed light on the optimization and generalization behavior of vastly overparameterized models (Pennington & Bahri, 2017; Wei et al., 2022), something that most traditional complexity measures could not (Zhang et al., 2021; Dziugaite et al., 2020).

Using gradient descent on a least square problem, our parameter update can be written as

$$\theta_{t+1} - \theta^* = \left(\mathbf{I}_p - \eta\Phi^\top\Phi\right)(\theta_t - \theta^*) \tag{4}$$

for $\Phi : \mathbb{R}^{n \times p}$ the feature matrix and $\theta^*$ the solution found by gradient descent. When choosing the learning rate $\eta$ optimally, the convergence rate is expressed as $\frac{\lambda_{\max}/\lambda_{\min}^+ - 1}{\lambda_{\max}/\lambda_{\min}^+ + 1}$ for $\lambda_{\min}^+$, respectively $\lambda_{\max}$, the smallest, resp. largest, non-zero eigenvalue of $\Phi^\top\Phi$ (Nesterov, 2003).

When $\Phi$ has i.i.d entries with zero mean, unit variance and bounded 4th moment, as we will consider in the rest of this paper, $\frac{1}{n}\Phi^\top\Phi$ follows the Wishart distribution. In particular, we have analytical expressions for the asymptotic minimum and maximum eigenvalue (section 2.2). When $\lim_{p,n\to\infty} p/n = \rho$, we have $\lambda_{\max}/\lambda_{\min}^+ \to \left(\frac{1+\sqrt{\rho}}{1-\sqrt{\rho}}\right)^2 \geq 1$. This function has asymptotes at 1, for $p = o(n)$ or $n = o(p)$ and diverges when $n = p$. This peak-shaped function can also be observed empirically in more complex models such as kernel methods (Poggio et al., 2019) and neural networks in the limit of infinite width as they are in the Neural Tangent Kernel regime (Lee et al., 2017; Jacot et al., 2018).

However, when using TD(0), presented in section 2.1, the feature covariance matrix $\Phi^\top\Phi$ would be replaced by $\Phi^\top\Xi(\mathbf{I}_n - \gamma\mathbf{P}^\pi)\Phi$. While Xiao et al. (2021) derived worst-case bounds for overparameterized TD, their dependency of the overparameterization ratio $\rho = p/n$ is not obvious. In this work we chose to use tools for *random matrix theory* for modeling the behavior of TD. While we can model $\Phi$ as an i.i.d matrix, it is not obvious how to model $\Xi$ and $\mathbf{P}^\pi$ in a realistic manner that would still allow us to characterize the spectrum of the TD operator $\Phi^\top\Xi(\mathbf{I}_n - \gamma\mathbf{P}^\pi)\Phi$.

---

[2]and an additional light tail decay condition, weaker than the bounded 4-th moment one (Tao & Vu, 2012).
[3]both in probability and almost surely.

For general $\Xi$ and $\mathbf{P}^\pi$, no closed form solution for the spectrum exists. We believe this is one of the key reasons why overparameterization in Temporal Difference learning has not been studied extensively. Thus, finding good typical models for $\mathbf{P}^\pi$ is of utmost importance, and the next section will introduce a class of models for which the spectrum of TD can be computed.

## 3.2 $\mathbf{P}^\pi$ as the Markov transition matrix of a random graph

In RL, $\mathbf{P}^\pi$ is the Markov transition matrix of a random walk when sampling from $\pi$. We will assume in the rest of the paper that the Markov chain is irreducible and aperiodic so that $\mathbf{P}^\pi$ has a unique stationary distribution denoted $\mathbf{d}_\pi$.

As we wish to understand the expected behavior of the TD algorithm, we propose in this subsection a simple yet expressive model of $\mathbf{P}^\pi$ that will allow us to do so. As random graphs are used to model properties of typical graphs, a candidate for a model of $\mathbf{P}^\pi$ would be to take the Markov transition matrix of a random graph. However, random graphs, at least the Erdős-Rényi(-Gilbert) type ones, are not able natively to take into account the impact of $\pi$, most importantly the fact that some state-actions are visited more often than others under $\mathbf{d}_\pi$, the stationary distribution of $\pi$. In this subsection, we propose a simple model for $\mathbf{P}^\pi$, denoted by $\hat{\mathbf{P}}^\pi$, which is the Markov transition matrix of a $\mathbb{G}(n, \mu)$ graph *deformed* by $\mathbf{d}_\pi$.

Let us start with an undirected graph of the type $\mathbb{G}(n, \mu)$. To take into account the state-action visitation $\mathbf{d}_\pi$, we construct the diagonal matrix $\mathbf{D}_\pi$ so that $\mathbf{D}_\pi = \text{diag}(\mathbf{d}_\pi)$. Now, we define the deformed adjacency matrix $\mathbf{A}^\pi$ as $\mathbf{A}^\pi \triangleq n\mathbf{A}\mathbf{D}_\pi$ where $\mathbf{A}$ is the original adjacency matrix. When $\mathbf{d}_\pi$ is uniform, we revert back to the original case $\mathbf{A}^\pi = \mathbf{A}$. This construction keeps the same edges as in the original graph but re-weights them by how likely $\mathbf{d}_\pi$ would visit that state-action. We can informally write the Markov transition matrix associated with this deformed graph as[4] $\hat{\mathbf{P}}^\pi = \mathbf{1}_n\mathbf{d}_\pi^\top + \mathbf{X}^\pi$ where (i) $\mathbf{1}_n\mathbf{d}_\pi^\top$ is a deterministic rank one matrix associated to the eigenvalue 1 as $\mathbf{1}_n^\top\mathbf{d}_\pi = 1$ and (ii) $\mathbf{X}^\pi = \sqrt{\frac{n(1-\mu)}{\mu}}\frac{1}{\sqrt{n}}\mathbf{W}\mathbf{D}_\pi$ is a stochastic matrix of null expectation as $\frac{1}{\sqrt{n}}\mathbf{W}$ is a Wigner-type matrix with zero mean and unit variance. It can be tempting to understand under which conditions the stochastic part becomes negligible and what it entails. As we will see shortly, this setting will reveal itself to be of importance when studying the spectrum of TD.

**Definition 1** (Asymptotically well-connected graphs). *We say a graph $\mathbb{G}$ is asymptotically well-connected if $|\lambda_2| \in o(1)$ for $1 = \lambda_1 \geq |\lambda_2| \geq \cdots \geq |\lambda_n|$ the (modulus) ordered eigenvalues of its Markov transition matrix.*

The choice for the name of Definition 1 comes from the fact that the second smallest eigenvalue of the Laplacian is related to the presence of *bottlenecks* in the graph through Cheeger's inequality (Cheeger, 1970).

*Example.* Graphs of the type $\mathbb{G}(n, \mu)$ (resp. $\mathbb{G}(n, d)$) are asymptotically well-connected if their expected degree $d = n\mu$ (resp. $d$) grows to $+\infty$ with $n$, i.e $1 \in o(d)$ (Zhao, 2012; Tran et al., 2010).

**Proposition 3.1.** *For the $\mathbf{d}_\pi$-deformed $\mathbb{G}(n, \mu)$ graph studied above, assuming $\hat{\mathbf{P}}^\pi = \mathbf{1}_n\mathbf{d}_\pi^\top + \mathbf{X}^\pi$, then the graph is asymptotically well-connected if $n\|\mathbf{d}_\pi\|_\infty \in o(\sqrt{d})$.*

This again generalizes the uniform case as, when $\mathbf{d}_\pi$ is uniform, the condition becomes equivalent to $1 \in o(d)$ as in the example. In the non-uniform case, this condition entails that either $\mathbf{d}_\pi$ spreads efficiently across states, or the degree $d$ rises fast enough that those states are still visited sufficiently often. Note that the *asymptotically well-connected* condition is also satisfied if we use longer back-ups of length $k$ such that $k$ grows with $n$ to $+\infty$, i.e $1 \in o(k)$ for the TD update. See Lemma B.2 for details.

In those cases, we have

$$\boxed{\hat{\mathbf{P}}^\pi = \mathbf{1}_n\mathbf{d}_\pi^\top + o(1)} \tag{5}$$

This approximation is particularly interesting as it asymptotically conserves the left and right eigenvectors, $\mathbf{d}_\pi$ and $\mathbf{1}_n$, of the original $\mathbf{P}^\pi$ associated to the eigenvalue 1. Furthermore, if all state-actions are reachable, the sequence of matrix power iterates of $\mathbf{P}^\pi$, $(\mathbf{P}^\pi)^k$ converges to $\mathbf{1}_n\mathbf{d}_\pi^\top$.

---

[4]Derivation can be found in Appendix.

While we are not the first to use tools from graph theory in the context of RL, related works we are aware of (Mahadevan & Maggioni, 2007; Machado et al., 2017, 2018; Wu et al., 2018) take a different approach and leverage eigenvectors of the Markov transition matrix (or Laplacian) to construct better representations for the value function or use them as options. In contrast, we use spectral properties of $\mathbf{P}^\pi$ to study the convergence of Temporal Difference learning.

### 3.3 Spiked Wishart model for TD with uniform sampling

Now that we have a simple random graph model for $\mathbf{P}^\pi$, we focus on estimating the spectrum of $\Phi^\top \Xi (\mathbf{I}_n - \gamma \hat{\mathbf{P}}^\pi) \Phi$. For simplicity, we assume the off-policy sampling is done uniformly, i.e $\Xi = \frac{1}{n} \mathbf{I}_n$. While this assumption is not realistic in most RL settings, it is still of interest in the context of dynamic programming where we have access to all the states.

**Proposition 3.2** (Spiked MP). *If $\rho \gamma^2 < 1$ the spectrum of $\frac{1}{n} \Phi^\top (\mathbf{I}_n - \gamma \hat{\mathbf{P}}^\pi) \Phi$ converges to the Marchenko-Pastur law with parameter $\rho$. When $\rho \gamma^2 \geq 1$, there is a phase transition (Baik et al., 2005) for the minimum non-zero eigenvalue $\lambda^+_{\min}$ which separates from bulk and converges to*

$$\lambda^+_{\min} \xrightarrow[p/n \to \rho]{a.s} \lambda^{spiked} = (1-\gamma)(\rho - \tfrac{1}{\gamma}) \tag{6}$$

Note that the phase transition is smooth as for $\rho = 1/\gamma^2$, $\lambda^{spiked} = (\sqrt{1/\gamma^2} - 1)^2$.

**Vast overparameterization and tabular case:** An interesting regime is when $\rho \gg \frac{1}{\gamma^2} > 1$. In that case, the maximum eigenvalue of the MP law and the spiked one differ by a factor $H = \frac{1}{1-\gamma}$, the *effective horizon* of the problem, as $\lim_{\rho \to \infty} \lambda_{\max}/\lambda^+_{\min} = \frac{1}{1-\gamma}$. In the tabular regime, $\Phi = \mathbf{I}_n$. As $\mathbf{P}^\pi$ has eigenvalues in $[-1, 1]$, the eigenvalues of $\mathbf{I}_n - \gamma \mathbf{P}^\pi$ are in $[1-\gamma, 1+\gamma]$. This would give a worst case ratio of $\frac{1+\gamma}{1-\gamma}$ which, up to a factor $1 + \gamma \leq 2$, is the same as the vastly overparameterized case above. As these eigenvalues govern the optimization speed of TD, the vastly overparameterized case and the tabular can be thought of as comparable.

### 3.4 Non-uniform off-policy sampling

In the previous subsection we were able to characterize the spectrum of $\Phi^\top \Xi (\mathbf{I}_n - \gamma \hat{\mathbf{P}}^\pi) \Phi$ assuming $\Xi = \frac{1}{n} \mathbf{I}_n$. In this subsection we analyze the more general case where $\Xi$ is not uniform. However by doing so, we lose guarantees and must resort to approximations. First, we analyze the eigenvalues of $\Xi (\mathbf{I}_n - \gamma \hat{\mathbf{P}}^\pi)$, or rather, in the limit of large $n$, the eigenvalues of $\Xi (\mathbf{I}_n - \gamma \mathbf{1}_n \mathbf{d}_\pi^\top)$.

**Lemma 3.1.** *If $\Xi = diag(\xi_1, \ldots, \xi_n)$ is non-singular, the eigenvalues of $\Xi (\mathbf{I}_n - \gamma \mathbf{1}_n \mathbf{d}_\pi^\top)$ satisfy a* secular *equation (Golub, 1973)*

$$1 - \gamma \sum_{i=1}^n \frac{\xi_i \cdot \mathbf{d}_\pi(i)}{\xi_i - \lambda} = 0 \tag{7}$$

This expression is another form of the *characteristic polynomial* of a matrix whose roots are its eigenvalues. However, this form enables us to derive approximations more easily. In particular, for the least visited state-action $m = \arg\min_{i=1,\ldots,n} \xi_i$, if $\xi_m \ll \xi_i, i \neq m$, we have for $\lambda \approx \xi_m$ that $\frac{1}{\xi_i - \lambda} \approx \frac{1}{\xi_i}$ if $i \neq m$. From this assumption, we derive an approximation for the minimum eigenvalue of $\Xi (\mathbf{I}_n - \gamma \hat{\mathbf{P}}^\pi)$

$$\ell_{\min} \approx \xi_m \frac{1-\gamma}{1-\gamma+\gamma\, \mathbf{d}_\pi(m)} \tag{8}$$

This eigenvalue is inferior to $\xi_m$ and reflects the interaction between the off-policy distribution and the discount factor. At worst, if $\mathbf{d}_\pi$ mostly visits state $m$, it will be close to $\xi_m(1-\gamma)$ which is the product of the minimum eigenvalues of $\Xi$ and $\mathbf{I}_n - \gamma \hat{\mathbf{P}}^\pi$. A similar expression can be derived for the maximum eigenvalue under similar assumptions.

Now, again reasoning approximately, if that eigenvalue was very small compared to the other ones who would form a bulk of similar values, we could expect $\Phi^\top \Xi (\mathbf{I}_n - \gamma \hat{\mathbf{P}}^\pi) \Phi$ to have a similar spiking behavior as before. However, in that case, the spiking eigenvalue would be

$$\lambda^+_{\min} \approx n\ell_{\min}\big(\rho - \tfrac{1}{1-n\ell_{\min}}\big) \tag{9}$$

While less simple than the estimators developed in the last subsection, we will see in Section 5 that it still accurately predicts the smallest non-zero eigenvalue.

# 4 Optimizing with Temporal Difference learning

## 4.1 Decomposition of the error

Using results about the spectrum of the TD operator shown in the last section, we analyze the behavior of the Mean Square Bellman Error $\mathcal{L}_{MSBE}(\mathbf{q}_t) = \frac{1}{2}\|\mathbf{r} + \gamma\mathbf{P}^\pi\mathbf{q}_t - \mathbf{q}_t\|_\Xi^2$ during the optimization. In particular, assuming $\Phi^\top\Xi(\mathbf{I}_n - \gamma\mathbf{P}^\pi)\Phi$ is non-singular for $p < n$

$$\mathbf{q}_{\text{TD}} = \begin{cases} \Phi\theta_{\text{TD}} = \Phi(\Phi^\top\Xi(\mathbf{I}_n - \gamma\mathbf{P}^\pi)\Phi)^{-1}\Phi^\top\Xi\mathbf{r}, \text{ if } p < n \\ \mathbf{q}^\pi \quad = \quad (\mathbf{I}_n - \gamma\mathbf{P}^\pi)^{-1}\mathbf{r}, \text{ if } p > n \end{cases}$$

the $\mathbf{q}$-value asymptotically reached by TD(0), we can decompose this error into two components.

**Lemma 4.1** (Decomposition of the error). *For $\mathbf{q}_t = \Phi\theta_t$, where $\theta_t$ is updated with TD(0) (eq. (2)), we have the following decomposition*

$$\|\mathbf{r} + \gamma\mathbf{P}^\pi\mathbf{q}_t - \mathbf{q}_t\|_\Xi \leq \sqrt{n\|\Xi\|_\infty} \cdot \|\mathbf{q}_t - \mathbf{q}_{TD}\|_2 + \|\mathbf{r} + \gamma\mathbf{P}^\pi\mathbf{q}_{TD} - \mathbf{q}_{TD}\|_\Xi \qquad (10)$$

*where $\|\mathbf{x}\|_\Xi = \sqrt{\mathbf{x}^\top\Xi\mathbf{x}}$ is the 2-norm in metric $\Xi$.*

Following Bottou & Bousquet (2007), we refer to $\|\mathbf{q}_t - \mathbf{q}_{\text{TD}}\|_2$ as the approximation error and $\|\mathbf{r} + \gamma\mathbf{P}^\pi\mathbf{q}_{\text{TD}} - \mathbf{q}_{\text{TD}}\|_\Xi$ as the estimation error. The factor $\sqrt{n\|\Xi\|_\infty} \geq 1$ comes from bounding the spectral norm of $(\mathbf{I}_n - \gamma\mathbf{P}^\pi)$ in metric $\Xi$. Beyond this factor, we are not exactly in the same setting as (Bottou & Bousquet, 2007) as the asymptotic TD(0) value estimate $\mathbf{q}_{\text{TD}}$ may not be the one reaching the lowest estimation error, as known in the RL community (Scherrer, 2010). Indeed, TD(0) finds the minimum of the Mean Square Projected Bellman Error (MSPBE) but not of the MSBE (Sutton & Barto, 2018). However in practice it is common to minimize the MSBE as it does not require an expensive projection step.

## 4.2 The ordinary behavior of the approximation error

In this subsection, we take a look at the approximation error, i.e how efficiently we converge to $\mathbf{q}_{\text{TD}}$.

**Proposition 4.1.** *Let us assume that $\Phi^\top\Xi(\mathbf{I}_n - \gamma\mathbf{P}^\pi)\Phi$ is diagonalizable as $\mathbf{Q}\Lambda\mathbf{Q}^{-1}$, and that its spectrum is real and positive. Denoting by $\lambda_{\min}^+$ and $\lambda_{\max}$ its smallest non-zero and largest eigenvalues, for $\eta = \frac{2}{\lambda_{\min}^+ + \lambda_{\max}}$*

$$\|\mathbf{q}_t - \mathbf{q}_{TD}\|_2^2 \leq \left(\frac{\lambda_{\max}/\lambda_{\min}^+ - 1}{\lambda_{\max}/\lambda_{\min}^+ + 1}\right)^{2t} K^2\|\mathbf{q}_0 - \mathbf{q}_{TD}\|_2^2 \qquad (11)$$

*where $K = \kappa(\Phi)\kappa(\mathbf{Q})$ when $p < n$ and $K = \kappa(\mathbf{Q})$ when $p > n$ where $\kappa$ is the condition number.*

The assumptions above can seem strong however they are verified under our model (cf Section 3.3). This is a slightly more general version of the linear convergence of gradient descent, however, as the TD operator may not be diagonalizable in an orthonormal basis, we incur an additional factor $K^2$.

## 4.3 The singular behavior of the estimation error

Now we turn our attention to the second term; the estimation error which is the *residual error* after $T \to +\infty$ steps. In supervised learning, we expect this error to be the usual bias of our algorithm and thus to decrease as $p/n$ increases until $p = n$ for which the bias would reach 0 as it is possible to interpolate the data. However, in reinforcement learning, the behavior of this term can be quite different, in particular it can be non-monotonous.

**Proposition 4.2.** *Let us assume $\hat{\mathbf{P}}^\pi$ satisfies the well-connected property, i.e that it is asymptotically rank one*

$$\lim_{\substack{n,p\to\infty \\ p/n\to\rho}} \|(\mathbf{I}_n - \gamma\hat{\mathbf{P}}^\pi)\mathbf{q}_{TD} - \mathbf{r}\|_\Xi^2 = \|\Pi_\perp\mathbf{r}\|_\Xi^2 + \frac{2\gamma}{1-\gamma\tilde{\rho}}(\mathbf{d}_\pi^\top\Pi\mathbf{r})(\xi^\top\Pi_\perp\mathbf{r}) + (\frac{\gamma}{1-\gamma\tilde{\rho}})^2\xi^\top\Pi_\perp\mathbf{1}_n(\mathbf{d}_\pi^\top\Pi\mathbf{r})^2$$

*where $\Pi$ is the orthogonal projector onto the span of $\Phi$ in norm $\Xi$: $\Pi = \Phi(\Phi^\top \Xi \Phi)^{-1} \Phi^\top \Xi$, $\Pi_\perp = \mathbf{I}_n - \Pi$ is its complement, and $\tilde{\rho} = \mathbf{d}_\pi^\top \Pi \mathbf{1}_n$.*

When $\gamma = 0$, i.e we just perform reward regression, this is equal to $\|\Pi_\perp \mathbf{r}\|_\Xi^2$ which is exactly the residual error in $\mathbf{r}$ that our representation cannot capture. When $\rho > 1$, $\Pi_\perp = 0$ so the limit is zero as we expect from a model able to interpolate the data. However, there is a behavior here not present in supervised learning. When $\tilde{\rho} \approx 1$, which we can expect when $\Pi$ is closer to identity, i.e $p$ close to but less than $n$, a *peaking* behavior of the estimation error can happen.

Replacing $\Pi$ by its expectation[5] $\rho \mathbf{I}_n$ above can lead to an equation simpler to analyze, albeit biased

$$1_{]0,1]}(\rho)\,(1-\rho)\big(\|\mathbf{r}\|_\Xi^2 + 2\frac{\gamma\rho}{1-\gamma\rho}(\mathbf{d}_\pi^\top \mathbf{r})(\xi^\top \mathbf{r}) + (\mathbf{d}_\pi^\top \mathbf{r})^2(\frac{\gamma\rho}{1-\gamma\rho})^2\big) \tag{12}$$

This phenomenon might appear counter-intuitive as we would expected our error to decrease with $p$. We provide a few insights to make sense of this. First, when $\gamma$ increases, so does the scale of the Q-function, so we can expect our errors to get bigger. Second, using random features $\Phi$ informally has a *smoothing* effect, stronger when $\rho$ is small, which causes the reduction in magnitude from $\frac{1}{1-\gamma}$ to $\frac{1}{1-\gamma\rho}$. This can be observed in the proof in appendix. Finally, peaking behaviors of TD such as this one have been observed in the community (Scherrer, 2010) but not theoretically explained in a general setting as far as we know.

## 4.4 Beyond stationary sampling

Until now, we always mentioned $n$ as the number of state-action pairs with the assumption that $\Xi$ was non-singular, i.e that our off-policy sampling *visited all the state-actions*. Here, we take a closer look at what happens when this is not the case. In particular, we study a simple toy model where we assume $\Xi$ only visits $n_{\text{vis}}(t)$ states at time $t$ with uniform probability. This simply changes all the $n$ by $n_{\text{vis}}(t)$ in the previous results. As often observed during training, at $t = 0$, $\pi$ might start as a more exploratory policy ($n_{\text{vis}}(t=0)$ high) and over time converges on a more deterministic solution ($n_{\text{vis}}(t)$ low). This can be problematic when performing TD learning with a fixed stepsize $\eta$. In that case, under the spiked model developed in Section 3.3 we need to ensure $\eta \leq 2/\big(1+\sqrt{p/n_{\text{vis}}(t)}\big)^2 \to 2/\lambda_{\max}$ or equivalently

$$n_{\text{vis}}(t) \geq p/\big(\sqrt{\tfrac{2}{\eta}}-1\big)^2 \tag{13}$$

As $n_{\text{vis}}(t)$ can typically decrease, **this condition can be broken and thus TD learning will diverge**. We conjecture this phenomenon may be related to the popularity of adaptive step size methods in RL over fixed step size ones.

## 5 Experiments

In all the experiments we use $\Phi$ i.i.d with normal entries. Per the universality property, our theoretical results hold for all distributions satisfying the hypotheses in Section 2.2.

**Study of the spectral distribution:** On Figure 1, we validate the theory and approximations of Section 3. For an asymmetric random graph $n = 4000, \mu = 1/2$, the spectrum of $\mathbf{P}$ (a) has an eigenvalue at 1 and the rest of order $1/\sqrt{n}$ (Bordenave et al., 2012). As predicted, the spectrum of the TD operator matches closely a spiked MP distribution (b). When the graph does not satisfy the well-connected property, for instance on a 4 rooms domain ($n = 4096$, $\pi$ stochastic[6]) (c) the spectral distribution is more complex (d). The relative error on $\lambda_{\min}^+$ is respectively 0.3% and 2%. Despite this, we show that the empirical and theoretical (eq. (9)) values for $\lambda_{\min}^+$ and $\lambda_{\max}$ on the 4 rooms domain ($n = 256$) closely match, both for $\Xi$ uniform or highly ill-conditioned with evenly spaced $\xi_i$ so that $\frac{\max_i \xi_i}{\min_i \xi_i} = 2000$. While (b) and (d) show how violating the *asymptotically well-connected* property breaks our spiked MP distribution result, we show on (e) and (f) that the extreme eigenvalues, which are the ones that matter for our optimization results, are still well predicted, even in more complex settings.

---

[5]when $\Xi = \frac{1}{n}\mathbf{I}_n$, by rotational invariance of $\Phi$, we $\mathbb{E}[\Pi] = \rho\mathbf{I}_n$, thus $\mathbb{E}[\tilde{\rho}] = \rho$.
[6]We sampled $\pi(a|s) \sim \text{Uniform}[0,1]$ and renormalized.

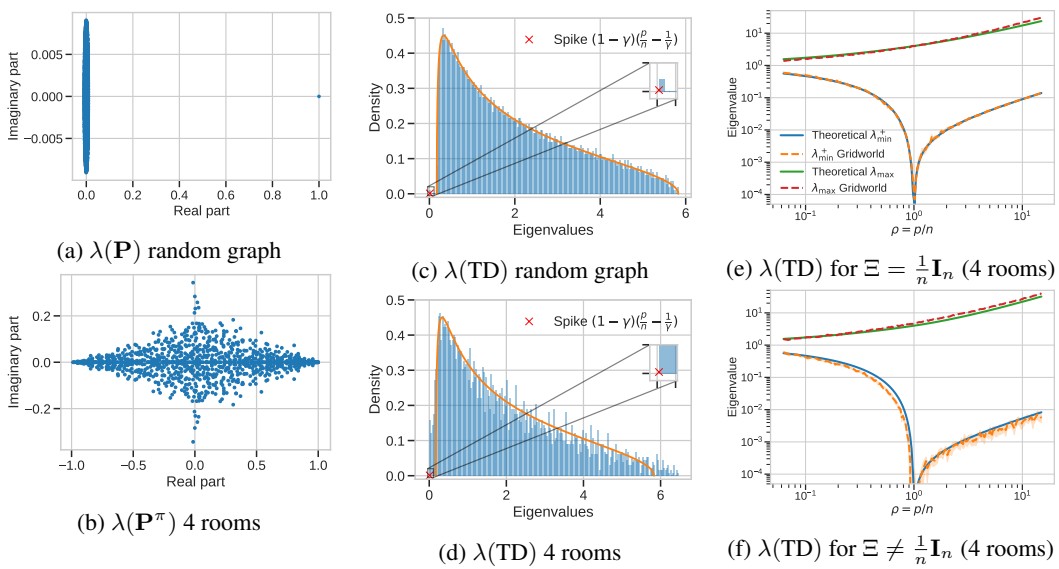

(a) $\lambda(\mathbf{P})$ random graph

(c) $\lambda(\mathrm{TD})$ random graph

(e) $\lambda(\mathrm{TD})$ for $\Xi = \frac{1}{n}\mathbf{I}_n$ (4 rooms)

(b) $\lambda(\mathbf{P}^\pi)$ 4 rooms

(d) $\lambda(\mathrm{TD})$ 4 rooms

(f) $\lambda(\mathrm{TD})$ for $\Xi \neq \frac{1}{n}\mathbf{I}_n$ (4 rooms)

Figure 1: (a) and (b): scatterplot of the spectra of the Markov transition matrix for a $\mathbb{G}(n,\mu)$ graph and the 4 rooms domain. (c) and (d): spectral distribution for the TD operator for $\rho = 2$. When the well-connected assumption is verified (a), the spectral distribution is a spiked ($\times$) Marchenko-Pastur (MP) law (in orange) (c). When the assumption is violated and $\hat{\mathbf{P}}^\pi \not\approx \mathbf{1}_n \mathbf{d}_\pi^\top$ (b), the distribution bleeds out of the MP support, yet the minimum eigenvalue is still close to the spike (d). On (e) and (f) we compare our prediction of the extreme eigenvalues $\lambda_{\min}^+$ and $\lambda_{\max}$ of the TD operator on the 4 rooms domain (Sutton et al., 1999) (b) as a function of $p/n$. We plot the empirical extreme eigenvalues (dotted) vs (e) MP theoretical ones (eqs. 3, 6) for $\Xi$ uniform and (f) approximation in eq. (9) for $\Xi$ non-uniform. Shaded areas are a 95% confidence interval when randomizing $\Phi$ and $\pi$ (10 seeds).

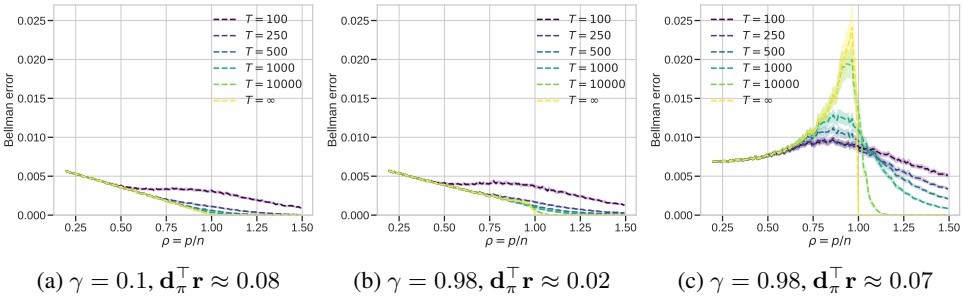

(a) $\gamma = 0.1$, $\mathbf{d}_\pi^\top \mathbf{r} \approx 0.08$

(b) $\gamma = 0.98$, $\mathbf{d}_\pi^\top \mathbf{r} \approx 0.02$

(c) $\gamma = 0.98$, $\mathbf{d}_\pi^\top \mathbf{r} \approx 0.07$

Figure 2: Value of the MSBE as a function of the ratio $p/n$ for various iterations $T$ during optimization. $T = +\infty$ corresponds to the asymptotic solution found by TD(0) (eq. (10)). We use uniform sampling ($\Xi = \frac{1}{n}\mathbf{I}_n$). On (a), for low $\gamma$ we observe a behavior similar to supervised learning where the residual error decreases and reaches 0 at $p = n$. (b) displays a similar behavior as the term $\mathbf{d}_\pi^\top \mathbf{r}$ is small. (c) As $\mathbf{d}_\pi^\top \mathbf{r}$ is higher, the *peaking* terms $\frac{1}{1-\gamma\bar{\rho}}$ are non-negligible. Shaded areas are a 99% confidence interval over randomness of $\theta_0$ and $\Phi$ (100 seeds).

**Optimization behavior of TD:** On Figure 2 we discuss the results of Section 4. We use again a 4 rooms domain with a fixed stochastic policy and study the Bellman error during training. On all subfigures of Figure 2, we observe that for $\rho \gg 1$ or $\rho \ll 1$, the optimization is faster as the curves for low $T$ are much closer to the asymptotic solution $T = +\infty$ (computed analytically) than for $\rho \approx 1$. This is in line with the result of section 4.2 for a spiked model: the ratio of eigenvalues which governs the optimization speed is $\lambda_{\max}/\lambda_{\min}^+ \to 1$ for $\rho \ll 1$ and to $\frac{1}{1-\gamma}$ for $\rho \gg 1$ while it diverges for $\rho \to 1$. This leads to a classical *double descent* phenomenon on (a)

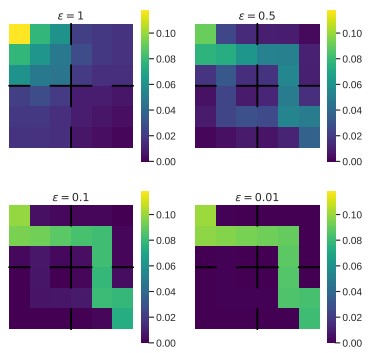

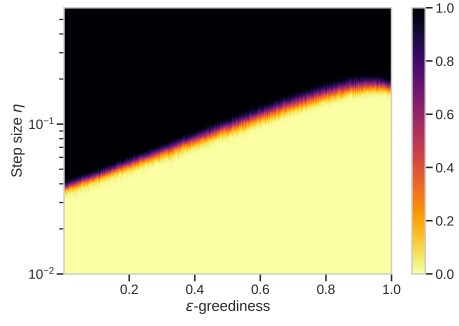

(a) $\mathbf{d}_\pi$ on 4 rooms for different $\epsilon$.

(b) Probability of divergence given step size and $\epsilon$-greediness.

Figure 3: (a): Example of the discounted state distribution in a small 4 rooms domain for varying $\epsilon$-greedy policies. (b) The probability of divergence of TD as a function of the step size $\eta$ and the $\epsilon$-greediness. For each point we report the probability it leads to divergence computed over 100 random $\Phi$: from yellow (0%) to black (100%).

and (b) purely due to optimization as predicted by Kuzborskij et al. (2021). On (a) and (b) because either $\gamma$ or $\mathbf{d}_\pi^\top \mathbf{r}$ is small, the terms of order $\frac{1}{1-\gamma\bar{\rho}}$ are negligible and thus it behaves similarly to supervised learning. On (c), the term $\frac{1}{1-\gamma\bar{\rho}}$ dominates and we observe the peaking behavior of the estimation error (Section 4.3). Furthermore on Figure 4 in appendix we study how our model and its approximation of Section 3.3 match the observed peaking behavior.

**Stability and non-stationarity:** On Figure 3 we highlight how $\Xi$ can affect the convergence of TD ($p = 200$). For this, we use a small 4 rooms ($n = 144$) environment[7] and compute an optimal deterministic policy $\pi^*$. Then, by varying the $\epsilon$ of a $\epsilon$-greedy policy based on $\pi^*$, we can analyze which combination of step size $\eta$ and $\epsilon$ would lead to convergence. On (a), we show the discounted state distribution for various $\epsilon$ where larger $\epsilon$ lead to more exploration. While our theoretical argument in Section 4.4 was about the number of visited states, we take here a more general and realistic approach where the state visitation is controlled by the greediness of our policy. Typically, during the optimization of a value based method, $\epsilon$ is annealed to 0 and the policy becomes more deterministic. On (b) we show the probability of diverging (averaged over 100 samples of $\Phi$) measured by computing the spectral radius of our iteration matrix. As predicted, for a fixed $\eta$, when $\epsilon$ decreases, we might go through a phase transition where the TD iteration, stable at the beginning, becomes unstable.

## 6   Conclusion

In this work, we have analyzed the spectral distribution of the Temporal Difference operator when using random features and modeling the Markov transition matrix as the one of a random graph. This characterization allowed us to make predictive theories for the optimization error and stability of TD. Notably we are able to predict how fast we converge and whether the solution found is accurate as a function of the ratio between the number of parameters and the number of state-action pairs. In particular, we highlighted and theoretically explained several phenomena specific to reinforcement learning such as the peaking behavior of the estimation error of TD or how changing the policy might cause divergence when using a fixed step size, even in the *on-policy* setting.

Important directions for future work include: (i) improving the model for $\mathbf{P}^\pi$ through careful analysis of more expressive random graphs, (ii) derive non-asymptotic results using the fluctuations of the extreme eigenvalues of the MP law (Baik et al., 2005) (iii) extend our results to neural networks in the Neural Tangent Kernel regime and finally (iv) study the behavior of policy optimization algorithms.

---

[7]where the bottom right state with a reward of 1 leads back to the initial state on the top left.

## Acknowledgments

We thank Alexandre Piché for useful comments on an earlier version of this paper.

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
