# A  Appendix: Experiments

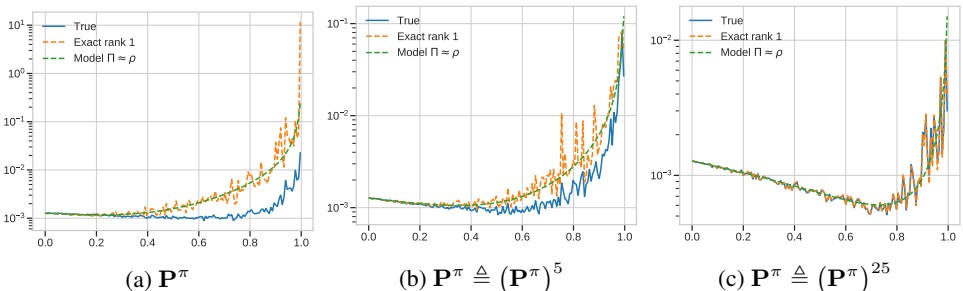

(a) $\mathbf{P}^\pi$  (b) $\mathbf{P}^\pi \triangleq \left(\mathbf{P}^\pi\right)^5$  (c) $\mathbf{P}^\pi \triangleq \left(\mathbf{P}^\pi\right)^{25}$

Figure 4: We plot the estimation error (in blue), the exact asymptotic model when $\hat{\mathbf{P}}^\pi$ becomes rank 1 (orange) in Section 4.3 and its simplification (green). On the $x$ axis we have the ratio $p/n$ and on the $y$ axis we have the value of the estimation error.

On Figure 4 we test the models developed in Section 4.3. Note that the models were developed for a $\hat{\mathbf{P}}^\pi$ approximately rank 1. We can see on (a) that the models, both the exact and the simplified over-estimate the true estimation error in a $14 \times 14$ 4 rooms domain ($n = 784$) for various $p = 1 \dots n$. However, if we were to do something akin to a k-step return, where we consider one action to be a sequence of k, this would be equivalent to changing $\hat{\mathbf{P}}^\pi$ to $\left(\hat{\mathbf{P}}^\pi\right)^k$. In (b) and (c) we can see that taking a higher k improve our estimation. Taking powers of the transition matrix has for effect to push down all the eigenvalues $< 1$ and as such make the graph "better" connected. All in all, while our formula is not precise in a realistic scenario, when considering an iterated setting, it recovers the estimation precisely.

Note that for Figure 4 we chose not to average over random $\Phi$ as it highlights the exactness of the rank one model developed in Section 4.3.

# B  Appendix: Proofs

In the paper and in the proofs, we extend naturally the notation $o(\cdot)$ to matrices with the following definition

$$\mathbf{A} \in o(1) \iff \|\mathbf{A}\| \in o(\|\mathbf{1}\|)$$

for $\|\cdot\|$ a natural operator norm. In this paper we only consider the 2-operator norm which corresponds to the largest singular value.

## B.1  Deformed random graph

In Section 2.3 we derive informally an expression for the Markov transition matrix of a random graph taking into account $\mathbf{d}_\pi$. Let us go into more details here, and explain why we called it an *informal* derivation. Let us recall that $\mathbf{A}^\pi = n\mathbf{A}\mathbf{D}_\pi$ for $\mathbf{A}$ the original adjacency matrix and $\mathbf{D}_\pi = \texttt{diag}(\mathbf{d}_\pi)$ the stationary distribution of $\pi$. To get $\hat{\mathbf{P}}^\pi$ we need to normalize the rows $i = 1, \dots n$ of $\mathbf{A}^\pi$ by their sum, i.e the degree $d_i$ of each state.

In the uniform case, $\mathbf{A}^\pi = \mathbf{A}$, (Bordenave et al., 2012) provide first a sketch of proof (eq 1.6) based on the fact that $d_i = \mathbf{D}_{ii} = \sum_j \mathbf{A}_{ij}$ satisfies $|d_i - n\mu| \in o(n)$ by the law of large numbers. This allows to "replace" $\mathbf{D}$ by $n\mu\mathbf{I}_n$ in the proofs which ties it back to the general Wigner/i.i.d case.

In the deformed case, we cannot invoke directly the law of large numbers. Indeed, now $d_i = \sum_j \mathbf{A}^\pi_{ij} = n\sum_j \mathbf{A}_{ij}\mathbf{d}_\pi(j)$ so we have a weighted sum instead of a simple one.

We need a concentration inequality for the weighted sum of random variables. One can be found in (Boucheron et al., 2013)

**Lemma B.1** (From (Boucheron et al., 2013)). *Let $X_1, X_2, \ldots, X_n$ be iid copies of a real random variable $X$ that obeys, for some $p \geq 1$*

$$\mathbb{P}(|X| > u) \leq \exp(-u^p)$$

*for all $u > 0$. Then there exists $L > 0$ such that, for any $s \in \mathbb{R}^n$, and for $q$ the conjugate of $p$ ($\frac{1}{p} + \frac{1}{q} = 1$)*

$$\mathbb{P}\left(\sum_{i=1}^n s_i X_i > t\right) \leq L \exp\left(-\frac{1}{L} \min\{\frac{t^p}{\|s\|_p^q}, \frac{t^2}{\|s\|_2^2}\}\right)$$

In our case, for $X_j = c \cdot (\mathbf{A}_{ij} - \mu)$ (where $c$ is chosen so the the tail assumption is satisfied),

$$\mathbb{P}(\sum_j \mathbf{d}_\pi(j)(\mathbf{A}_{ij} - \mu) > t) \leq L \exp\left(-\frac{1}{L} \min\{\frac{(ct)^p}{\|\mathbf{d}_\pi\|_p^q}, \frac{(ct)^2}{\|\mathbf{d}_\pi\|_2^2}\}\right)$$

$$\mathbb{P}(\frac{d_i}{n} - \mu > t) \leq L \exp\left(-\frac{1}{L} \min\{\frac{(ct)^p}{\|\mathbf{d}_\pi\|_p^q}, \frac{(ct)^2}{\|\mathbf{d}_\pi\|_2^2}\}\right)$$

Thus, if $\|\mathbf{d}_\pi\|_p^q$ and $\|\mathbf{d}_\pi\|_2^2$ are $\in o(1)$, and then reasoning on $-X_j$ we have that $|\frac{d_i}{n} - \mu|$ is, almost surely when $n$ is large, smaller than any constant $t > 0$, therefore $o(1)$. Thus we get the same argument $\mathbf{D} = n\mu\mathbf{I}_n + o(n)$. Note here that in high dimension the stronger condition for $\mathbf{d}_\pi$ will be for $p = \infty$, $\|\mathbf{d}_\pi\|_\infty \in o(1)$. As we already make a stronger assumption later, we have chosen not to include it in the main text.

All of this allows to calculate $\hat{\mathbf{P}}^\pi$ by normalizing $\mathbf{A}^\pi$

$$\hat{\mathbf{P}}^\pi = \mathbf{D}^{-1}\mathbf{A}^\pi$$
$$\approx (\frac{1}{n\mu})n\mathbf{A}\mathbf{D}_\pi$$
$$= (\frac{1}{\mu})(\mu\mathbf{1}_n\mathbf{1}_n^\top + \sigma\mathbf{W})\mathbf{D}_\pi$$
$$= \mathbf{1}_n\mathbf{d}_\pi^\top + \frac{\sigma}{\mu}\mathbf{W}\mathbf{D}_\pi$$

When the entries of $\mathbf{A}$ are iid Bernoulli of mean $\mu$, we have $\sigma^2 = \mu(1 - \mu)$ and $\mathbf{W}$ has iid whitened Bernoulli entries.

We consider this proof informal as there are a certain number of technicalities involved when controlling the deviation of all these random quantities (as we have sketched above for the expected degree).

## B.2   Asymptotically well-connected graph

**Proposition 3.1.** *For the $\mathbf{d}_\pi$-deformed $\mathbb{G}(n, \mu)$ graph studied above, assuming $\hat{\mathbf{P}}^\pi = \mathbf{1}_n\mathbf{d}_\pi^\top + \mathbf{X}^\pi$, then the graph is asymptotically well-connected if $n\|\mathbf{d}_\pi\|_\infty \in o(\sqrt{d})$.*

*Proof.* First, recall that

$$\mathbf{X}^\pi = \sqrt{\frac{n(1 - \mu)}{\mu}} \frac{1}{\sqrt{n}}\mathbf{W}\mathbf{D}_\pi$$

For $\|\cdot\|$ the 2-operator corresponding to the maximum singular value, we have:

$$\|\mathbf{X}^\pi\| \leq \sqrt{\frac{n(1 - \mu)}{\mu}} \|\frac{1}{\sqrt{n}}\mathbf{W}\|\|\mathbf{D}_\pi\|$$

From (Tao, 2012) we have for both Wigner (Corollary 2.3.6), when the graph is undirected and i.i.d matrices (Corollary 2.3.5) when the edges are iid and directed, that There exists $A, C, c > 0$

$$\mathbb{P}(\|\tfrac{1}{\sqrt{n}}\mathbf{W}\| \geq A) \leq C \exp(-cn)$$

Which can be understood as $\tfrac{1}{\sqrt{n}}\mathbf{W}$ has bounded operator norm with overwhelming probability. In particular, taking the limit $n \to \infty$ gives us that $\tfrac{1}{\sqrt{n}}\mathbf{W}$ has bounded norm almost surely when $n \to \infty$.

Then $\|\mathbf{D}_\pi\| = \|\mathbf{d}_\pi\|_\infty$ where $\|\cdot\|_\infty$ is the vector max-norm. This is due to the fact that $\mathbf{D}_\pi$ is a diagonal matrix with positive entries, therefore its largest singular value is its largest eigenvalue.

For $\|\mathbf{X}^\pi\|$ to be $o(1)$ we need

$$\sqrt{\frac{n(1-\mu)}{\mu}} \cdot \|\mathbf{d}_\pi\|_\infty \cdot \|\tfrac{1}{\sqrt{n}}\mathbf{W}\| \in o(1)$$

As $1 - \mu \leq 1$ and for $n\mu = d$, it is sufficient to have (almost surely)

$$n\|\mathbf{d}_\pi\|_\infty \in o(\sqrt{d})$$

$\square$

**Lemma B.2.** *From (Broder & Karlin, 1989), for a irreducible and aperiodic Markov transition matrix $\mathbf{P}^\pi$, we have that, for $\left(\mathbf{P}_{ij}^\pi\right)^k = \mathbf{d}_{\pi j} + O(\sqrt{\frac{\mathbf{d}_{\pi j}}{\mathbf{d}_{\pi i}}}\lambda_2^k)$, thus*

$$\left(\mathbf{P}^\pi\right)^k = \mathbf{1}\mathbf{d}_\pi^\top + O(\sqrt{\kappa(\mathbf{d}_\pi)}\lambda_2^k)$$

*where $\kappa(\cdot)$ is the condition number.*

This lemma means that when using longer back-ups, as is for instance done when using n-step returns, the second eigenvalue of $\mathbf{P}^k$ will shrink to $0$ as long as $1 \in o(k)$.

### B.3    Spiked Marchenko-Pastur distribution

For, we prove small result showing that the $o(1)$ appearing in $\hat{\mathbf{P}}^\pi$ remains a $o(1)$ in $\tfrac{1}{n}\Phi^\top(\mathbf{I}_n - \gamma\hat{\mathbf{P}}^\pi)\Phi$.

**Lemma B.3.** *For $\hat{\mathbf{P}}^\pi$ satisfying the asymptotically well-connected condition, we have*

$$\frac{1}{n}\Phi^\top(\mathbf{I}_n - \gamma\hat{\mathbf{P}}^\pi)\Phi = \frac{1}{n}\Phi^\top(\mathbf{I}_n - \gamma\mathbf{1}_n\mathbf{d}_\pi^\top)\Phi + o(1)$$

*Proof.* For $\|\cdot\| = \|\cdot\|_2$ the 2-operator norm which corresponds to the largest singular value, we have

$$\frac{1}{n}\|\Phi^\top o(1)\Phi\| \leq \frac{1}{n}\|\Phi\|^2 \cdot \|o(1)\|$$

$$= \frac{1}{n}\left(\sqrt{\lambda_{\max}(\Phi^\top\Phi)}\right)^2 o(1)$$

As the maximum eigenvalue of $\tfrac{1}{n}\Phi^\top\Phi$ converges a.s to $(1+\sqrt{\rho})^2$ per the Marchenko-Pastur theorem

$$\frac{1}{n}\|\Phi^\top o(1)\Phi\| \xrightarrow{a.s} o(1)$$

$\square$

This lemma is a useful result to understand that a $o(1)$ perturbation stays a $o(1)$ perturbation when adding features.

**Proposition B.1** (Asymmetric rank 1 perturbation). *We consider the spectrum of $\frac{1}{n}\Phi^\top(\mathbf{I}_n - \gamma\mathbf{u}\mathbf{v}^\top)\Phi$ for $\mathbf{u}, \mathbf{v} \in \mathbb{R}^n$ so that $\mathbf{v}^\top\mathbf{u} = 1$ and $\gamma \in [0,1[$ which is a rank 1 perturbation of the Wishart covariance model. If $\rho\gamma^2 < 1$ the spectrum of $\frac{1}{n}\Phi^\top(\mathbf{I}_n - \gamma\mathbf{u}\mathbf{v}^\top)\Phi$ converges to the Marchenko-Pastur law. When $\rho\gamma^2 \geq 1$, the empirical spectral distribution also converges to the Marchenko-Pastur law, but additionally, we observe a phase transition (Baik et al., 2005), and an eigenvalue $\lambda^{spiked}$ separates from the bulk*

$$\boxed{\lambda^+_{\min} \xrightarrow[p/n\to\rho]{a.s} \lambda^{spiked} = (1-\gamma)(\rho - \tfrac{1}{\gamma})} \tag{14}$$

*Proof.* The proof is based on a simple adaptation of the proof in (Couillet & Liao, 2021) (page 117). As we consider an asymmetric spike, we lose commutative properties. Thus, we can only prove our result in the case where there is a unique spike instead of a finite number as is usual. We are looking to characterize the eigenvalues of $\frac{1}{n}\Phi^\top(\mathbf{I}_n - \gamma\mathbf{u}\mathbf{v}^\top)\Phi$, or equivalently (up to some zeros), the eigenvalues of $\frac{1}{n}\Phi\Phi^\top(\mathbf{I}_n - \gamma\mathbf{u}\mathbf{v}^\top)$. However, in this case, $\frac{1}{n}\Phi\Phi^\top$ is not a sample from the Wishart distribution, but $\frac{1}{p}\Phi\Phi^\top$ is. We therefore analyze $\frac{1}{p}\Phi\Phi^\top$ and will simply have to scale our spectrum by $\rho$ at the end.

$$0 = \det\left(\frac{1}{p}\Phi\Phi^\top(\mathbf{I}_n - \gamma\mathbf{u}\mathbf{v}^\top) - \hat{\lambda}\mathbf{I}_n\right)$$

$$0 = \det\left(\frac{1}{p}\Phi\Phi^\top - \hat{\lambda}(\mathbf{I}_n - \gamma\mathbf{u}\mathbf{v}^\top)^{-1}\right)\det(\mathbf{I}_n - \gamma\mathbf{u}\mathbf{v}^\top)$$

Using Sherman-Morrison formula, we have $(\mathbf{I}_n - \gamma\mathbf{u}\mathbf{v}^\top)^{-1} = \mathbf{I}_n + \frac{\gamma}{1-\gamma}\mathbf{u}\mathbf{v}^\top$ as $\mathbf{v}^\top\mathbf{u} = 1$.

Since $\det\left(\mathbf{I}_n - \gamma\mathbf{u}\mathbf{v}^\top)\right) \neq 0$, the second determinant can be discarded. Then, we can factor the resolvent of the Wishart model. That is, letting $\mathbf{Q}(\hat{\lambda}) = \left(\frac{1}{p}\Phi\Phi^\top - \hat{\lambda}\mathbf{I}_n\right)^{-1}$, we write

$$0 = \det\left(\tfrac{1}{p}\Phi\Phi^\top - \hat{\lambda}\mathbf{I}_n - \hat{\lambda}\tfrac{\gamma}{1-\gamma}\mathbf{u}\mathbf{v}^\top\right)$$

$$= \det\mathbf{Q}^{-1}(\hat{\lambda})\det\left(\mathbf{I}_n - \hat{\lambda}\mathbf{Q}(\hat{\lambda})\tfrac{\gamma}{1-\gamma}\mathbf{u}\mathbf{v}^\top\right).$$

Inverting $\frac{1}{p}\Phi\Phi^\top - \hat{\lambda}\mathbf{I}_n$ is almost surely possible when $\hat{\lambda} < (1 - \sqrt{1/\rho})^2$ per the concentration of the empirical spectral distribution of Wishart covariance models to the Marchenko-Pastur law and the term $\mathbf{Q}^{-1}(\hat{\lambda})$ is almost surely non-zero. So the left term can be discarded. For the second we use Sylvester's determinant identity to end up with a scalar equation

$$0 = \left(1 - \hat{\lambda}\mathbf{v}^\top\mathbf{Q}(\hat{\lambda})\mathbf{u}\tfrac{\gamma}{1-\gamma}\right).$$

As $\mathbf{Q}(\hat{\lambda})$ is a deterministic equivalent of $m(\hat{\lambda})\mathbf{I}_n$ ((Marčenko & Pastur, 1967), Theorem 2.4 in (Couillet & Liao, 2021)), where $m(\cdot)$ is the Stieljes transform of the MP distribution, thus

$$\hat{\lambda}\mathbf{v}^\top\mathbf{Q}(\hat{\lambda})\mathbf{u}\frac{\gamma}{1-\gamma} = \hat{\lambda}m(\hat{\lambda})\frac{\gamma}{1-\gamma} + o(1)$$

The above determinant is zero when

$$\hat{\lambda}m(\hat{\lambda}) = \frac{1-\gamma}{\gamma} + o(1) \tag{15}$$

$\square$

From the definition of $m(\cdot)$ as the Stieljes transform of the MP distribution, we know (Couillet & Liao, 2021) it satisfies

$$zm(z) = -1 + \frac{1}{1 - z - czm(z)}$$

for $c = \lim n, p \to \infty \, {}^n/_p = {}^1/_\rho$ Thus, for $z = \hat{\lambda}$

$$\hat{\lambda} = 1 - c\hat{\lambda}m(\hat{\lambda}) - \frac{1}{1 + \hat{\lambda}m(\hat{\lambda})}$$

Plugging the equation above, asymptotically

$$\hat{\lambda} = 1 - c\frac{1-\gamma}{\gamma} - \frac{1}{1 + \frac{1-\gamma}{\gamma}}$$

$$= 1 - c\frac{1-\gamma}{\gamma} - \gamma$$

$$= (1-\gamma)(1 - \frac{1}{\rho\gamma})$$

Now, we have to remember to multiply our spectrum by $\rho$ to obtain the original one, and so we obtain

$$\boxed{\hat{\lambda} = (1-\gamma)(\rho - \frac{1}{\gamma})}$$

When $(1-\gamma)(\rho - \frac{1}{\gamma}) \leq (1 - \sqrt{\rho})^2$ which is equivalent to $\rho\gamma^2 \geq 1$, this eigenvalue is isolated on the left of the bulk.

**Proposition 3.2** (Spiked MP). *If $\rho\gamma^2 < 1$ the spectrum of $\frac{1}{n}\Phi^\top(\mathbf{I}_n - \gamma\hat{\mathbf{P}}^\pi)\Phi$ converges to the Marchenko-Pastur law with parameter $\rho$. When $\rho\gamma^2 \geq 1$, there is a phase transition (Baik et al., 2005) for the minimum non-zero eigenvalue $\lambda_{\min}^+$ which separates from bulk and converges to*

$$\boxed{\lambda_{\min}^+ \xrightarrow[p/n\to\rho]{a.s} \lambda^{spiked} = (1-\gamma)(\rho - \frac{1}{\gamma})} \tag{6}$$

*Proof.* We essentially do the same proof as above, but with an additional $o(1)$ moving around and we show that it does not impact the Stieljes transform, thus the spectrum, almost surely.

$$0 = \det\left(\frac{1}{n}\Phi\Phi^\top(\mathbf{I}_n - \gamma\hat{\mathbf{P}}^\pi) - \hat{\lambda}\mathbf{I}_n\right)$$

$$0 = \det\left(\frac{1}{n}\Phi\Phi^\top(\mathbf{I}_n - \gamma\mathbf{1}_n\mathbf{d}_\pi^\top + o(1)) - \hat{\lambda}\mathbf{I}_n\right)$$

Where we used the definition of $\hat{\mathbf{P}}^\pi$ and Lemma B.3. Then

$$0 = \det\left(\frac{1}{n}\Phi\Phi^\top - \hat{\lambda}(\mathbf{I}_n - \gamma\mathbf{1}_n\mathbf{d}_\pi^\top + o(1))^{-1}\right)\det(\mathbf{I}_n - \gamma\mathbf{1}_n\mathbf{d}_\pi^\top + o(1))$$

As the eigenvalues of $\mathbf{I}_n - \gamma\mathbf{1}_n\mathbf{d}_\pi^\top$ are lower bounded by $1-\gamma$ for $n$ large enough $\det(\mathbf{I}_n - \gamma\mathbf{1}_n\mathbf{d}_\pi^\top + o(1))$ must be non-zero. As $(\mathbf{I}_n + o(1))^{-1} = \sum_i (o(1))^i = \mathbf{I}_n + o(1)$, using Sherman-Morrison formula, we have $(\mathbf{I}_n + o(1) - \gamma\mathbf{1}_n\mathbf{d}_\pi^\top)^{-1} = \mathbf{I}_n + o(1) + o(1)\frac{\gamma(1+o(1))}{1-\gamma+o(1)}\mathbf{1}_n\mathbf{d}_\pi^\top = \mathbf{I}_n + \frac{\gamma}{1-\gamma}\mathbf{1}_n\mathbf{d}_\pi^\top + o(1)$ as $\mathbf{d}_\pi^\top\mathbf{1}_n = 1$

With the same derivation and arguments as above

$$0 = \det\left(\frac{1}{n}\Phi\Phi^\top - \hat{\lambda}\mathbf{I}_n - \hat{\lambda}\frac{\gamma}{1-\gamma}\mathbf{1}_n\mathbf{d}_\pi^\top + o(1)\right)$$

$$= \det\mathbf{Q}^{-1}(\hat{\lambda})\det\left(\mathbf{I}_n - \hat{\lambda}\mathbf{Q}(\hat{\lambda})\left(\frac{\gamma}{1-\gamma}\mathbf{1}_n\mathbf{d}_\pi^\top + o(1)\right)\right).$$

And finally we end up with the same equality as before where the $o(1)$ from the perturbation and the one from the deterministic equivalent fuse together. As the determinant is continuous

$$\hat{\lambda}\mathbf{d}_\pi^\top\mathbf{Q}(\hat{\lambda})\mathbf{1}_n\frac{\gamma}{1-\gamma} = \hat{\lambda}m(\hat{\lambda})\frac{\gamma}{1-\gamma} + o(1)$$

Therefore the Stieljes transforms when using $\hat{\mathbf{P}}^\pi$ and $\mathbf{1}_n\mathbf{d}_\pi^\top$ are asymptotically the same, thus, almost surely, they have the same empirical spectral distribution.

$\square$

## B.4   Approximation of the extreme eigenvalues

**Lemma 3.1.** *If $\Xi = \mathtt{diag}(\xi_1, \ldots, \xi_n)$ is non-singular, the eigenvalues of $\Xi(\mathbf{I}_n - \gamma\mathbf{1}_n\mathbf{d}_\pi^\top)$ satisfy a secular equation (Golub, 1973)*

$$1 - \gamma \sum_{i=1}^{n} \frac{\xi_i \cdot \mathbf{d}_\pi(i)}{\xi_i - \lambda} = 0 \tag{7}$$

*Proof.* Let us consider $\mathbf{v} \neq 0$ and $\lambda$, an eigenvector of $\Xi(\mathbf{I}_n - \gamma\mathbf{1}_n\mathbf{d}_\pi^\top)$ and its associated eigenvalue.

$$\lambda\mathbf{v} = \Xi(\mathbf{I}_n - \gamma\mathbf{1}_n\mathbf{d}_\pi^\top)\mathbf{v}$$

This give on component $i$

$$\lambda v_i = \xi_i v_i - \gamma\xi_i \sum_j \mathbf{d}_\pi(j)v_j$$

Then

$$\gamma\xi_i \sum_j \mathbf{d}_\pi(j)v_j = (\xi_i - \lambda)v_i$$

Assuming $\lambda \neq \xi_i, \forall i$

$$\gamma\frac{\xi_i \sum_j \mathbf{d}_\pi(j)v_j}{\xi_i - \lambda} = v_i$$

Multiplying by $\mathbf{d}_\pi(i)$ and summing

$$\gamma \sum_i \frac{\mathbf{d}_\pi(i)\xi_i \cdot \sum_j \mathbf{d}_\pi(j)v_j}{\xi_i - \lambda} = \sum_i \mathbf{d}_\pi(i)v_i$$

Now the same term $\sum_j \mathbf{d}_\pi(j)v_j = \mathbf{d}_\pi^\top\mathbf{v}$ appears on both sides. Let us verify if it can be null. If that were the case, we would have $\Xi(\mathbf{I}_n - \gamma\mathbf{1}_n\mathbf{d}_\pi^\top)\mathbf{v} = \Xi\mathbf{v} = \lambda\mathbf{v}$. Thus, as $\Xi$ is diagonal, the possible vectors $\mathbf{v}$ possible are the canonical basis vectors and the possible eigenvalues are $\xi_i$. However, as $\mathbf{d}_\pi$ is a positive vector whose entries sum to 1, we cannot have $\mathbf{d}_\pi^\top\mathbf{v} = 0$. Thus $\mathbf{d}_\pi^\top\mathbf{v} \neq 0$ and by dividing on each side:

$$\gamma \sum_i \frac{\xi_i\mathbf{d}_\pi(i)}{\xi_i - \lambda} = 1$$

$\square$

## B.5   Optimization and approximation error

**Lemma 4.1** (Decomposition of the error). *For $\mathbf{q}_t = \Phi\theta_t$, where $\theta_t$ is updated with TD(0) (eq. (2)), we have the following decomposition*

$$\|\mathbf{r} + \gamma\mathbf{P}^\pi\mathbf{q}_t - \mathbf{q}_t\|_\Xi \leq \sqrt{n\|\Xi\|_\infty} \cdot \|\mathbf{q}_t - \mathbf{q}_{TD}\|_2 + \|\mathbf{r} + \gamma\mathbf{P}^\pi\mathbf{q}_{TD} - \mathbf{q}_{TD}\|_\Xi \tag{10}$$

*where $\|\mathbf{x}\|_\Xi = \sqrt{\mathbf{x}^\top\Xi\mathbf{x}}$ is the 2-norm in metric $\Xi$.*

*Proof.*

$$\|\mathbf{r} + \gamma \mathbf{P}^\pi \mathbf{q}_t - \mathbf{q}_t\|_\Xi = \|\gamma \mathbf{P}^\pi(\mathbf{q}_t - \mathbf{q}_{\text{TD}}) - (\mathbf{q}_t - \mathbf{q}_{\text{TD}}) + \mathbf{r} + \gamma \mathbf{P}^\pi \mathbf{q}_{\text{TD}} - \mathbf{q}_{\text{TD}}\|_\Xi$$

$$\leq \|\gamma \mathbf{P}^\pi(\mathbf{q}_t - \mathbf{q}_{\text{TD}}) - (\mathbf{q}_t - \mathbf{q}_{\text{TD}})\|_\Xi + \|\mathbf{r} + \gamma \mathbf{P}^\pi \mathbf{q}_{\text{TD}} - \mathbf{q}_{\text{TD}}\|_\Xi$$

$$= \sqrt{(\mathbf{q}_t - \mathbf{q}_{\text{TD}})^\top (\mathbf{I}_n - \gamma \mathbf{P}^\pi)^\top \Xi (\mathbf{I}_n - \gamma \mathbf{P}^\pi)(\mathbf{q}_t - \mathbf{q}_{\text{TD}})} + \|\mathbf{r} + \gamma \mathbf{P}^\pi \mathbf{q}_{\text{TD}} - \mathbf{q}_{\text{TD}}\|_\Xi$$

$$\leq \|\sqrt{\Xi}(\mathbf{I}_n - \gamma \mathbf{P}^\pi)\|_2 \|\mathbf{q}_t - \mathbf{q}_{\text{TD}}\|_2 + \|\mathbf{r} + \gamma \mathbf{P}^\pi \mathbf{q}_{\text{TD}} - \mathbf{q}_{\text{TD}}\|_\Xi$$

$$\leq \|\sqrt{\Xi}\|_2 \cdot \|\mathbf{I}_n - \gamma \mathbf{P}^\pi\|_2 \|\mathbf{q}_t - \mathbf{q}_{\text{TD}}\|_2 + \|\mathbf{r} + \gamma \mathbf{P}^\pi \mathbf{q}_{\text{TD}} - \mathbf{q}_{\text{TD}}\|_\Xi$$

$$\leq \sqrt{\|\xi\|_\infty} \sqrt{n} \|\mathbf{I}_n - \gamma \mathbf{P}^\pi\|_\infty \|\mathbf{q}_t - \mathbf{q}_{\text{TD}}\|_2 + \|\mathbf{r} + \gamma \mathbf{P}^\pi \mathbf{q}_{\text{TD}} - \mathbf{q}_{\text{TD}}\|_\Xi$$

$$\leq \sqrt{\|\xi\|_\infty} \sqrt{n} \|\mathbf{q}_t - \mathbf{q}_{\text{TD}}\|_2 + \|\mathbf{r} + \gamma \mathbf{P}^\pi \mathbf{q}_{\text{TD}} - \mathbf{q}_{\text{TD}}\|_\Xi$$

Where we used $\|\mathbf{A}\|_2 \leq \sqrt{n}\|\mathbf{A}\|_\infty$ and the fact that $\mathbf{P}^\pi$ is a stochastic matrix, thus has entries between 0 and 1. As such, the entries of $\mathbf{I}_n - \gamma \mathbf{P}^\pi$ are between $-\gamma$ and 1, in particular they are bounded by 1. $\qquad\square$

**Proposition 4.1.** *Let us assume that $\Phi^\top \Xi (\mathbf{I}_n - \gamma \mathbf{P}^\pi)\Phi$ is diagonalizable as $\mathbf{Q}\Lambda\mathbf{Q}^{-1}$, and that its spectrum is real and positive. Denoting by $\lambda_{\min}^+$ and $\lambda_{\max}$ its smallest non-zero and largest eigenvalues, for $\eta = \frac{2}{\lambda_{\min}^+ + \lambda_{\max}}$*

$$\|\mathbf{q}_t - \mathbf{q}_{TD}\|_2^2 \leq \left(\frac{\lambda_{\max}/\lambda_{\min}^+ - 1}{\lambda_{\max}/\lambda_{\min}^+ + 1}\right)^{2t} K^2 \|\mathbf{q}_0 - \mathbf{q}_{TD}\|_2^2 \tag{11}$$

*where $K = \kappa(\Phi)\kappa(\mathbf{Q})$ when $p < n$ and $K = \kappa(\mathbf{Q})$ when $p > n$ where $\kappa$ is the condition number.*

*Proof.* Let us begin with the case $p > n$ From the TD iteration Equation (2) multiplied on the left by $\Phi$

$$\mathbf{q}_{t+1} = \Phi\Phi^\top \Xi \mathbf{r} - \Phi\Phi^\top \Xi (\mathbf{I}_n - \gamma \mathbf{P}^\pi)\mathbf{q}_t$$

For a i.i.d matrix $\Phi$, we have that $\Phi\Phi^\top$ is almost surely invertible. Thus the fixed point of this iteration is

$$\mathbf{q}_{\text{TD}} = \left(\Phi\Phi^\top \Xi (\mathbf{I}_n - \gamma \mathbf{P}^\pi)\right)^{-1} \Phi\Phi^\top \Xi \mathbf{r}$$

$$= \left(\mathbf{I}_n - \gamma \mathbf{P}^\pi\right)^{-1} \Xi^{-1} (\Phi\Phi^\top)^{-1} \Phi\Phi^\top \Xi \mathbf{r}$$

$$= \left(\mathbf{I}_n - \gamma \mathbf{P}^\pi\right)^{-1} \mathbf{r}$$

$$= \mathbf{q}^\pi$$

Thus, for $\mathbf{H}_{\text{TD}} = \Phi^\top \Xi (\mathbf{I}_n - \gamma \mathbf{P}^\pi)\Phi$ and $\mathbf{x}_t = \mathbf{q}_t - \mathbf{q}^\pi$

$$\|\mathbf{x}_t\|_2^2 = \|(\mathbf{I}_n - \eta\mathbf{H}_{\text{TD}})^t \mathbf{x}_0\|_2^2$$

$$\leq \|(\mathbf{I}_n - \mathbf{H}_{\text{TD}})^t\|_2^2 \|\mathbf{x}_0\|_2^2$$

$$= \|\mathbf{Q}(\mathbf{I}_n - \eta\Lambda)^t \mathbf{Q}^{-1}\|_2^2 \|\mathbf{x}_0\|_2^2$$

$$\leq \|\mathbf{Q}\|_2^2 \|\mathbf{Q}^{-1}\|_2^2 \|(\mathbf{I}_n - \eta\Lambda)^t\|_2^2 \|\mathbf{x}_0\|_2^2$$

$$= \kappa(\mathbf{Q})^2 \max_{i,\lambda_i = \Lambda_{ii}} \{|1 - \eta\lambda_i|\}^{2t} \|\mathbf{x}_0\|_2^2$$

As $\mathbf{I}_n - \eta\Lambda$ is diagonal, and as such its largest singular value (2-norm) is its largest eigenvalue (in modulus).

Now, we supposed the eigenvalues were real, $\max_{i,\lambda_i=\Lambda_{ii}}\{|1-\eta\lambda_i|\}$ will be minimized when the largest and smallest terms are equal.

$$|1-\eta\lambda_{\min}^+| = |1-\eta\lambda_{\max}|$$

If $\lambda_{\min}^+ = \lambda_{\max}$ this can be achieved for any $\eta$, however the only solution when they are different is to have the two expressions above of a different sign

$$1-\eta\lambda_{\min}^+ = -1 + \eta\lambda_{\max}$$

Which leads to $\eta = \frac{2}{\lambda_{\min}^+ + \lambda_{\max}}$ and

$$|1-\eta\lambda_{\min}^+| = |1-\eta\lambda_{\max}| = \frac{\lambda_{\max} - \lambda_{\min}^+}{\lambda_{\max} + \lambda_{\min}^+}$$

Now for $p < n$ we do a similar reasoning with $\theta$. By the exact same reasoning, but using $\mathbf{H}_{\mathrm{TD}} = \Phi^\top\Xi(\mathbf{I}_n - \gamma\mathbf{P}^\pi)\Phi$ instead and the fixed point $\theta_{\mathrm{TD}} = \left(\Phi^\top\Xi(\mathbf{I}_n - \gamma\mathbf{P}^\pi)\Phi\right)^1\Phi^\top\Xi\mathbf{r}$ we get

$$\|\theta_t - \theta_{\mathrm{TD}}\| \leq \kappa(\mathbf{Q})^2 \left(\frac{\lambda_{\max} - \lambda_{\min}^+}{\lambda_{\max} + \lambda_{\min}^+}\right)^{2t}\|\theta_0 - \theta_{\mathrm{TD}}\|_2^2$$

Now, we would like to obtain a result for the Q-function associated to these parameters. We have $\sigma_{\min}(\Phi)\|\mathbf{x}\|_2 \leq \|\Phi\mathbf{x}\|_2 \leq \sigma_{\max}(\Phi)\|\mathbf{x}\|_2$ for $\sigma_{\max}$ and $\sigma_{\min}$ the max and min singular values. Thus, multiplying each side by $\sigma_{\min}(\Phi)\sigma_{\max}(\Phi)$, we get

$$\sigma_{\min}(\Phi)\|\mathbf{q}_t - \mathbf{q}_{\mathrm{TD}}\| \leq \sigma_{\min}(\Phi)\sigma_{\max}(\Phi)\|\theta_t - \theta_{\mathrm{TD}}\|$$

and

$$\sigma_{\min}(\Phi)\sigma_{\max}(\Phi)\|\theta_0 - \theta_{\mathrm{TD}}\|_2 \leq \sigma_{\max}(\Phi)\|\mathbf{q}_0 - \mathbf{q}_{\mathrm{TD}}\|$$

Squaring these results and putting it back in the contraction equation, we get for $\mathbf{q}_{\mathrm{TD}} = \Phi\theta_{\mathrm{TD}}$

$$\|\mathbf{q}_t - \mathbf{q}_{\mathrm{TD}}\|^2 \leq \kappa(\mathbf{Q})^2 \left(\frac{\lambda_{\max} - \lambda_{\min}^+}{\lambda_{\max} + \lambda_{\min}^+}\right)^{2t} \left(\frac{\sigma_{\max}(\Phi)}{\sigma_{\min}(\Phi)}\right)^2\|\mathbf{q}_0 - \mathbf{q}_{\mathrm{TD}}\|_2^2$$

$$= \kappa(\mathbf{Q})^2 \left(\frac{\lambda_{\max} - \lambda_{\min}^+}{\lambda_{\max} + \lambda_{\min}^+}\right)^{2t} \kappa(\Phi)^2\|\mathbf{q}_0 - \mathbf{q}_{\mathrm{TD}}\|_2^2$$

$\square$

## B.6 Peaking behavior of the estimation error

**Proposition 4.2.** *Let us assume $\hat{\mathbf{P}}^\pi$ satisfies the well-connected property, i.e that it is asymptotically rank one*

$$\lim_{\substack{n,p\to\infty \\ p/n\to\rho}} \|(\mathbf{I}_n - \gamma\hat{\mathbf{P}}^\pi)\mathbf{q}_{TD} - \mathbf{r}\|_\Xi^2 = \|\Pi_\perp\mathbf{r}\|_\Xi^2 + \frac{2\gamma}{1-\gamma\tilde{\rho}}(\mathbf{d}_\pi^\top\Pi\mathbf{r})(\xi^\top\Pi_\perp\mathbf{r}) + \left(\frac{\gamma}{1-\gamma\tilde{\rho}}\right)^2\xi^\top\Pi_\perp\mathbf{1}_n(\mathbf{d}_\pi^\top\Pi\mathbf{r})^2$$

*where $\Pi$ is the orthogonal projector onto the span of $\Phi$ in norm $\Xi$: $\Pi = \Phi(\Phi^\top\Xi\Phi)^{-1}\Phi^\top\Xi$, $\Pi_\perp = \mathbf{I}_n - \Pi$ is its complement, and $\tilde{\rho} = \mathbf{d}_\pi^\top\Pi\mathbf{1}_n$.*

*Proof.* Let us look at the case $p < n$. We have $\mathbf{q}_{\mathrm{TD}} = \Phi\left(\Phi^\top\Xi(\mathbf{I} - \gamma\mathbf{P}^\pi)\Phi\right)^{-1}\Phi^\top\Xi\mathbf{r}$ We will therefore study more closely the matrix $\Phi\left(\Phi^\top\Xi(\mathbf{I} - \gamma\mathbf{P})\Phi\right)^{-1}\Phi^\top\Xi$, in the case where $\mathbf{P}^\pi = \mathbf{u}\mathbf{v}^\top$ is a rank one matrix. First, Sherman-Morrison formula gives us

$$(\mathbf{A} - \gamma\mathbf{x}\mathbf{y}^\top)^{-1} = \mathbf{A}^{-1} + \gamma\frac{\mathbf{A}^{-1}\mathbf{x}\mathbf{y}^\top\mathbf{A}^{-1}}{1-\gamma\mathbf{y}^\top\mathbf{A}^{-1}\mathbf{x}}$$

which is the usual formula but for $\mathbf{x} \triangleq -\gamma\mathbf{x}$. Note that this is licit as long as $1 - \gamma\mathbf{y}^\top\mathbf{A}^{-1}\mathbf{x} \neq 0$. Using Sherman-Morrison's formula

$$
\begin{aligned}
\Phi\big(\Phi^\top\Xi(\mathbf{I}-\gamma\mathbf{P}^\pi)\Phi\big)^{-1}\Phi^\top\Xi &= \Phi\big(\Phi^\top\Xi\Phi - \gamma\Phi^\top\Xi\mathbf{u}\mathbf{v}^\top\Phi\big)^{-1}\Phi^\top\Xi \\
&= \Phi\big(\Phi^\top\Xi\Phi - \gamma\Phi^\top\Xi\mathbf{u}(\Phi^\top\mathbf{v})^\top\big)^{-1}\Phi^\top\Xi \\
&= \Phi\Big((\Phi^\top\Xi\Phi)^{-1} + \gamma\frac{(\Phi^\top\Xi\Phi)^{-1}\Phi^\top\Xi\mathbf{u}(\Phi^\top\mathbf{v})^\top(\Phi^\top\Xi\Phi)^{-1}}{1 - \gamma\mathbf{v}^\top\Phi(\Phi^\top\Xi\Phi)^{-1}\Phi^\top\Xi\mathbf{u}}\Big)\Phi^\top\Xi \\
&= \Phi(\Phi^\top\Xi\Phi)^{-1}\Phi^\top\Xi + \gamma\frac{\Phi(\Phi^\top\Xi\Phi)^{-1}\Phi^\top\Xi\mathbf{u}\mathbf{v}^\top\Phi(\Phi^\top\Xi\Phi)^{-1}\Phi^\top\Xi}{1 - \gamma\mathbf{v}^\top\Phi(\Phi^\top\Xi\Phi)^{-1}\Phi^\top\Xi\mathbf{u}} \\
&= \Phi(\Phi^\top\Xi\Phi)^{-1}\Phi^\top\Xi + \gamma\frac{\Phi(\Phi^\top\Xi\Phi)^{-1}\Phi^\top\Xi\mathbf{u}\mathbf{v}^\top\Phi(\Phi^\top\Xi\Phi)^{-1}\Phi^\top\Xi}{1 - \gamma\mathbf{v}^\top\Phi(\Phi^\top\Xi\Phi)^{-1}\Phi^\top\Xi\mathbf{u}} \\
&= \Pi + \gamma\frac{\Pi\mathbf{u}\mathbf{v}^\top\Pi}{1 - \gamma\mathbf{v}^\top\Pi\mathbf{u}}
\end{aligned}
$$

For $\Pi = \Phi(\Phi^\top\Xi\Phi)^{-1}\Phi^\top\Xi$ the usual projector unto the span of $\Phi$ in metric $\Xi$.

Now, for the TD error we obtain

$$
\begin{aligned}
(\mathbf{I}-\gamma\mathbf{P}^\pi)\mathbf{q}_{\text{TD}} - \mathbf{r} &= (\mathbf{I}-\gamma\mathbf{P}^\pi)\big(\Pi + \gamma\frac{\Pi\mathbf{u}\mathbf{v}^\top\Pi}{1 - \gamma\mathbf{v}^\top\Pi\mathbf{u}}\big)\mathbf{r} - \mathbf{r} \\
&= \big(\Pi + \gamma\frac{\Pi\mathbf{u}\mathbf{v}^\top\Pi}{1 - \gamma\mathbf{v}^\top\Pi\mathbf{u}} - \gamma\mathbf{P}^\pi\Pi - \gamma\mathbf{u}\mathbf{v}^\top\gamma\frac{\Pi\mathbf{u}\mathbf{v}^\top\Pi}{1 - \gamma\mathbf{v}^\top\Pi\mathbf{u}}\big)\mathbf{r} - \mathbf{r} \\
&= \big(\Pi + \gamma\frac{\Pi\mathbf{u}\mathbf{v}^\top\Pi}{1 - \gamma\mathbf{v}^\top\Pi\mathbf{u}} - \gamma\mathbf{u}\mathbf{v}^\top\Pi - \gamma^2\mathbf{v}^\top\Pi\mathbf{u}\frac{\mathbf{u}\mathbf{v}^\top\Pi}{1 - \gamma\mathbf{v}^\top\Pi\mathbf{u}}\big)\mathbf{r} - \mathbf{r} \\
&= \big(\Pi + \gamma\frac{\Pi\mathbf{u}\mathbf{v}^\top\Pi}{1 - \gamma\mathbf{v}^\top\Pi\mathbf{u}} - \gamma\frac{\mathbf{u}\mathbf{v}^\top\Pi}{1 - \gamma\mathbf{v}^\top\Pi\mathbf{u}}\big)\mathbf{r} - \mathbf{r} \\
&= \big(\Pi - \mathbf{I} + \gamma\frac{(\Pi-\mathbf{I})\mathbf{u}\mathbf{v}^\top\Pi}{1 - \gamma\mathbf{v}^\top\Pi\mathbf{u}}\big)\mathbf{r} \\
&= (\Pi - \mathbf{I})\big(\mathbf{I} + \frac{\gamma}{1 - \gamma\mathbf{v}^\top\Pi\mathbf{u}}\mathbf{u}\mathbf{v}^\top\Pi\big)\mathbf{r}
\end{aligned}
$$

Let's call $\mathbf{I} - \Pi = \Pi_\perp$ the complement projector projector and $\tilde{\rho} = \mathbf{v}^\top\Pi\mathbf{u}$. Now, for the MSBE error, we get

$$
\begin{aligned}
\|(\mathbf{I}-\gamma\mathbf{P})\mathbf{q}_{\text{TD}} - \mathbf{r}\|_\Xi^2 &= \mathbf{r}^\top\big(\mathbf{I} + \frac{\gamma}{1-\gamma\tilde{\rho}}\mathbf{u}\mathbf{v}^\top\Pi\big)^\top(\Pi-\mathbf{I})^\top\Xi(\Pi-\mathbf{I})\big(\mathbf{I} + \frac{\gamma}{1-\gamma\tilde{\rho}}\mathbf{u}\mathbf{v}^\top\Pi\big)\mathbf{r} \\
&= \mathbf{r}^\top\big(\mathbf{I} + \frac{\gamma}{1-\gamma\tilde{\rho}}\Pi^\top\mathbf{v}\mathbf{u}^\top\big)\Pi_\perp^\top\Xi\Pi_\perp\big(\mathbf{I} + \frac{\gamma}{1-\gamma\tilde{\rho}}\mathbf{u}\mathbf{v}^\top\Pi\big)\mathbf{r} \\
&= \mathbf{r}^\top\big(\Pi_\perp^\top\Xi\Pi_\perp + \frac{\gamma}{1-\gamma\tilde{\rho}}\Pi^\top\mathbf{v}\mathbf{u}^\top\Xi\Pi_\perp + \Pi_\perp^\top\Xi\frac{\gamma}{1-\gamma\tilde{\rho}}\mathbf{u}\mathbf{v}^\top\Pi + \frac{\gamma}{1-\gamma\tilde{\rho}}\Pi^\top\mathbf{v}\mathbf{u}^\top\Xi\Pi_\perp\frac{\gamma}{1-\gamma\tilde{\rho}}\mathbf{u}\mathbf{v}^\top\Pi\big)\mathbf{r} \\
&= \|\Pi_\perp\mathbf{r}\|_\Xi^2 + \frac{2\gamma}{1-\gamma\tilde{\rho}}(\mathbf{v}^\top\Pi\mathbf{r})(\xi^\top\Pi_\perp\mathbf{r}) + \big(\frac{\gamma}{1-\gamma\tilde{\rho}}\big)^2(\mathbf{v}^\top\Pi\mathbf{r})^2(\xi^\top\Pi\mathbf{u})
\end{aligned}
$$

where we use the fact that $\Pi^\top\Xi\Pi = \Xi\Pi = \Pi^\top\Xi$ and same for $\Pi_\perp$.

Using $\mathbf{v} = \mathbf{d}_\pi$ and $\mathbf{u} = \mathbf{1}_n$ gives us the approximation.

$\qquad\qquad\qquad\qquad\qquad\qquad\qquad\qquad\qquad\qquad\qquad\qquad\qquad\qquad\qquad\qquad\square$

# C    Colab links

For plotting the spiked MP spectra `https://colab.research.google.com/drive/1E3DgmZQF1-1FNYeVo4msyGUHwRiK32WE?usp=sharing`

For the extreme eigenvalues approximation `https://colab.research.google.com/drive/1H_GkjzlGyUxB0i55jM5VAqLXoEGGxN3h?usp=sharing`

For the peaking behavior `https://colab.research.google.com/drive/1EVudyGfieXOUqDLifdhFrs8Pu1OzKOAW?usp=sharing`

For the peaking behavior model `https://colab.research.google.com/drive/1pLIqaV5BO1ChIj9y9UmP5jmlWx26FNI7?usp=sharing`

For the unstability wrt $\eta$ and $\epsilon$ `https://colab.research.google.com/drive/1CuPZhv3eZoINXHCKLtwjVlHrLy9ylBuI?usp=sharing`