# OpenReview forum: "On the role of overparameterization in off-policy Temporal Difference learning with linear function approximation"
_NeurIPS.cc/2022/Conference — NeurIPS 2022 Accept_

### Official Review · Reviewer_EBEx · 2022-07-10

**Rating:** 5
**Confidence:** 2
**Soundness:** 3 good
**Presentation:** 2 fair
**Contribution:** 2 fair

**Summary:**

This paper uses ideas from Random matrix theory to study the spectrum of the TD update operator under strong assumptions on the MDP graph. In doing so, the authors focus on the over-parameterized case where the number of parameters in the linear function approximation exceeds the total number of distinct (s, a) pairs.



**Questions:**

* Proposition 4.1 assumes that the spectrum of the relevant matrix is real and positive but this can be easily violated which is the main challenge in going from least squares to the TD version.

* In Proposition 4.2, again how can you justify assuming that the transition matrix is rank one just by taking the limit? This seems to be assuming away the main complexity of MDPs and lack of any structure in the graphs.

* There is a blurb about Wigner type matrices in Page 3, but I don't notice anything related in the rest of the paper.




**Limitations:**

Adequate

**Strengths And Weaknesses:**

Strengths:

* Paper is well motivated and proposes making use of results from random matrix theory to analyze the spectrum of the TD operator.

Weaknesses:

* Section 3.2 appears weak, and unfortunately the assumptions here are very strong and a critical part of the paper. The justification for modeling the $P^\pi$ as a modification of a random graph is very unclear to me (Section 3.2) and this seems to be a key ingredient of the entire analysis.

* Motivation for several of the assumptions is clearly tractability but unclear takeaways from the analysis. This includes assumptions like random features, rank one transition matrix. The former seems okay but the latter is too strong and the justifications for this are very hand-wavy and unclear.

* Analysis of value estimation for TD learning under linear approximation has also been considered by Xiao et. al at ICLR 2022: https://openreview.net/forum?id=shbAgEsk3qM which the authors should discuss.

---

> ### Author Response · Authors · 2022-08-02
> **Answer EBEx**
>
> We thank the review for his careful reading of the paper and the relevance of the points raised.
>
> ### Weaknesses
> 1 and 2. We have added an extensive discussion in the general message we highly encourage you to read. There we justify why this model is a good tradeoff between having an analytical solution and an expressive model, why even if the assumption is violated our model of the extreme eigenvalues is still accurate (Fig 1.e, 1.f) and how using $k$-step TD as often done in practice would be in line with this assumption.
>
> 3. We were not aware of this recent work, thank you for pointing it to us, we will update the paper accordingly after reading it carefully. While it has a different goal with a different approach, it is still very relevant and should be discussed.
>
> ### Questions
> 1. We believe this assumption is acceptable in light of Prop 3.2 where we show that under our assumptions the spectrum will indeed be a spiked MP distribution and thus real and positive asymptotically. Furthermore, the convergence of the iteration $\theta_{t+1} = \theta_{t} - \eta A \theta_t$ is well studied and known to converge (for $\eta$ small enough) when the spectrum of $A$ is included in the positive real part complex half-plane. See for instance https://arxiv.org/abs/2007.05520, more generally these are known as _stable_ matrices and the assumption of 4.1 can be extended to them, but it was not necessary to because of the result of 3.2.
> 2. Yes this is the main hypothesis and limitation of the paper, the asymptotically well-connected/rank 1 assumption. It is indeed a strong one as discussed above and in the general message. In spirit it is very much similar to a mean field method where we replace a complex system by a simplified version matching only some of its moments. In that regard our model is wrong, in the same way that Ising models or a large part of thermodynamics is wrong. However we believe our model is still useful and highly predictive of the extreme eigenvalues which are the important ones in optimization.
> 3. The Wigner matrix actually appears in the adjacency matrix of the random graph in section 3.2. It is mentioned on line 136, but the details of it appear mainly in the derivation in appendix B.1 and B.2 where we use the fact that (rescaled) Wigner matrices have a bounded operator norm with high probability.
>
> Thank you again for the review, we hope we addressed some of your concerns, do not hesitate to let us know how you think the paper could be further improved!

---

> > ### Author Response · Authors · 2022-08-09
> > **Related work**
> >
> > Concerning the paper you recommended to us, here is a more detailed comparison:
> >
> > - They derive worst-case bound for the convergence/generalization of TD (and related algorithms), some of which use a regularization (which we don't assume we do).
> > - Their bounds (eq 20-21) depends on the minimum eigenvalue of some matrix depending on the features and the off-policy distribution
> >
> > We think this is actually complementary to our work in the sense that we are mainly interested in characterizing the spectrum. This in turn could be used in their bounds to explicit how the off-policiness, overparameterization ratio and other quantities interplay in the optimization.
> > Furthermore, our extensive analysis of the spectrum allowed us to predict some RL specific behaviors (Fig 2 and fig 3) which we do not think could be predicted from their work.
> >
> > This is however very relevant and we changed slightly the intro section and the end of sec 3.1 to reflect their work.
> > While we would like to add some details in the paper, such as the ones above, we are tight on space but we will update it with a more in-depth discussion if the paper is accepted and we have an additional page.
> >
> > Thank you again for the recommendation, we hope our answers made our contribution clearer and shed light on why our assumption, despite being strong, is still highly predictive of the support of the spectrum (which is matters most when deriving bounds).

---

### Official Review · Reviewer_MGtX · 2022-07-12

**Rating:** 5
**Confidence:** 2
**Soundness:** 2 fair
**Presentation:** 1 poor
**Contribution:** 2 fair

**Summary:**

This paper studies the largely unexplored area of the overparameterization in RL and TD learning.

**Questions:**

Overall, the paper is well written and includes interesting results. Some comments to improve the paper are as follows. Please clarify them in the paper.
1) This paper addresses off-policy TD learning. It is not clear until eq (2). It would be better to put more emphasis on that the paper considers off-policy TD.
2) Eq (2) would be be TD-learning because it does not include any samples. It is a determininstic algorithm .
3) Needs some changes in expressions: e.g. "0 mean" can be replaced with "zero mean." What is the meaning of spectrum of TD?
4) What is the meaning of overparameterization in TD? Please clarify it in the paper.
5) Overall, the paper is written based on languages used in deep learning fields. It would be better to consider readability for people in RL areas.
6) More discussions are needed: why do we study spectrum of TD?

**Limitations:**

The work includes interesting results, but the writing should be more syetematic for general readers.

**Strengths And Weaknesses:**

Strengths:
This paper studies the largely unexplored area of the overparameterization in RL and TD learning.
It is a unique contribution, and the paper is well written.

Weaknesses:
The work includes interesting results, but the writing should be more syetematic for general readers.
The presentation is rather sloppy.

---

> ### Author Response · Authors · 2022-08-02
> **Answer MGtX**
>
> We thank you for your review and for appreciating the originality of our contribution!
> Writing a paper mixing random matrix theory, random graph theory and reinforcement learning was not an easy task, and we tried our best to present it in a format that would allow researchers from different fields to grasp our contribution. We understand this can still be improved and we welcome any practical suggestion you have in order to improve our paper.
>
> 1. Yes this is fair, although it was mentioned in our 1) contribution in the introduction, we added off-policy in the title of the paper and mentioned it in the abstract as well, as it might clarify our generality for readers who might think our contribution is limited to the on-policy setting.
> 2. We believe both versions would indeed be called TD learning however we indeed did not precise it was the expected version. We changed it to " algorithm TD(0) [8] whose expected update is" one line 61. Please let us know if this sounds more accurate.
> 3. We have changed the typos as suggested. By spectrum of TD we mean the spectrum of the TD operator $\Phi^\top \Xi (I_n - \gamma P^\pi) \Phi$ as described on line 114-116. We can include a sentence in the preliminaries about this notation if you think it would make it clearer for the reader.
> 4. Overparameterization would be when the number of parameters $p$ is larger than the size of the MDP $n$. This correspond to $\rho = \frac{p}{n} > 1$.
> 5. We are not sure how to address this comment. The main author(s) of the paper have an expertise in reinforcement learning and have chosen their notation to be consistent with some commonly used in RL theory (for instance Schocknecht 2002, https://arxiv.org/abs/1011.4362 or https://arxiv.org/abs/2007.05520). Furthermore we do not model anything beyond linear models in this paper. In order to be accessible to a wider audience, which parts of the paper would you recommend us to reformulate?
> 6. When using an operator to update our parameters, for instance $\Phi^\top \Xi (I_n - \gamma P^\pi) \Phi$ for TD or $\Phi^\top \Phi$ for an ordinary least squares model, understanding its spectrum is key to understanding the behavior of that operator and therefore of our algorithm. Studying the spectrum of TD allows us to make predictions about the optimization behavior of TD. Our model allowed us to predict and explain some phenomena specific to the RL setting (Fig 2 and 3) which, as far as we know, did not have an explanation before. Finally, a hope we would have for future work, is that when the link between model properties/architecture and optimization is well understood, this could give us some insight on which hyperparameters and architecture matter for optimization and how to select them before even training. This could reduce cost of training very large models (while unrelated to our approach, a theoretical approach was recently able to do so https://arxiv.org/abs/2203.03466)
>
> We would be happy to incorporate comments/ideas you have about the clarity of paper, thank you again for the review!

---

### Official Review · Reviewer_72Nh · 2022-07-14

**Rating:** 6
**Confidence:** 3
**Soundness:** 3 good
**Presentation:** 3 good
**Contribution:** 3 good

**Summary:**

This paper introduces a random graph model allowing to study the stability and optimization properties of the TD algorithm in RL. More specifically, this paper applies tools from random matrix theory to study the spectral distribution of the (projected) TD(0) operator and uses this understanding to study the behavior of the approximation and estimation error in different regimes of over- and under-parametrization as the size of the system diverges.

**Questions:**

My main question concerns the weakness highlighted above: I think the authors should stress in which sense the model they observe is a relevant one to the RL community, as else it might be perceived as being reverse-engineered.

If this is not possible this point should be discussed much more clearly in the conclusion as a limitation.

**Limitations:**

Depending on the answer to the previous question, the authors might have to address much more precisely the applicative limitations of the model being investigated.

**Strengths And Weaknesses:**

The paper is very well written, clearly introduces the reader to the fundamental concepts of random matrix theory that are needed to understand the results, and presents such results in a clear way. The paper addresses an important theoretical issue: understanding the role of overparametrization in RL, and does so, although under some relatively strong assumptions, discussing a nontrivial case and presenting some theoretical understanding of a previously unexplained phenomenon.. From a first reading of the supplementary material, the technical part of the paper seems sound. The numerics solidly corroborate the findings of the paper and hint to a new direction of research.

The main weakness of the paper is the significance: the model of random graph being investigated seems a bit artificial: it seems unclear from a first reading how many relevant problems in RL can be captured in this setting.

---

> ### Author Response · Authors · 2022-08-02
> **Answer 72Nh**
>
> We thank you for your review and for appreciating our contribution and its importance. Your point concerning the asymptotically well-connected / rank 1 model of the transition matrix is fair. As this point has been raised by other reviews we have dedicated an extensive discussion of it in the general message. In summary, the points we made were that: a) the model is limited but is a tradeoff between expressiveness/interpretability (through the closed form solutions) and complexity (while we use a simple model we are still have to fully take into account the effect of the stationary distribution). b) While the assumption is wrong in most settings, the model is still useful for predicting the extreme eigenvalues (the important ones). And finally, c) using $k$-step TD as often done in practice, for $k$ growing with $n$ (the MDP size) would actually lead to our assumption being true as the second eigenvalue would be contracted to the exponent $k$.
>
> Furthermore concerning the limitation of the rank 1 assumption, more can be seen on Fig 4 in the appendix. In this setting we compare the peak predicted by our model (prop 4.2) and a simplification of it and eq (12). As you can see, while the presence of the peak is predicted, its magnitude is not well captured for short traces. However when using longer traces ($k=25$) the model recovers very precisely the behavior of the optimization error, as predicted by our point c).
>
> While we are still very much limited by space, if you think the following discussion of the assumptions and their limitations is interesting, we will add it to the main text of the paper.
> Thank you again for your review and the suggestions!

---

### Official Review · Reviewer_3zWt · 2022-07-18

**Rating:** 9
**Confidence:** 5
**Soundness:** 4 excellent
**Presentation:** 4 excellent
**Contribution:** 4 excellent

**Summary:**

This paper investigate the role of optimization in value approximation with linear value function approximators leveraging tools from random matrix theory and random graph theory. The authors characterize the spectrum of the TD operator, and show that the resulting learning process demonstrates clear signatures of the double descent phase-transition near the critically parameterized cut-off.

**Questions:**

I think this paper is quite well-written as it is. I will update here if questions arise.

**Limitations:**

One of the issues that complicate the analysis of TD is the interplay between the sampling distribution, the transition matrix, and the function approximator itself. I think this work did a fantastic job.

It would obviously be beneficial to extend the analysis here to neural networks. However, I am not sure if the NTK regime is the best place as the authors indicated in the discussion because over-parameterized ReLU is just one way the network could be parameterized and it is known to contain bias towards low-complexity functions. Recent work [^2] [^3] have shown empirically how such issues could be addressed by modifying the parameterization itself.

Therefore instead of working with MLPs as they are, over-parameterized or not, the analysis offered here might just as well be abstract enough that it offers more insight than an analysis focused on particular types of overparameterized neural networks.

[^2]: Achiam, J., Knight, E. and Abbeel, P. (2019) ‘Towards Characterizing Divergence in Deep Q-Learning’, arXiv [cs.LG]. Available at: http://arxiv.org/abs/1903.08894.

[^3]: Yang, G., Ajay, A. and Agrawal, P. (2021) ‘Overcoming The Spectral Bias of Neural Value Approximation’, in International Conference on Learning Representations.

**Strengths And Weaknesses:**

The paper is well-written, and all of the proofs are correct as far as I understand. The result is relevant and is a necessary extension of the analysis on overparameterization in the supervised learning regime [^1].

A notable technical contribution is the introduction of a class of models for which the spectrum of TD can be computed. The experimental result is clearly presented and offers useful insight. The figures on the evolution of the maximum and minimum singular values, as well as the overall distribution in response to parameterization ratio and different types of graph clearly convey the main point of the paper.

[^1]: Kuzborskij and Szepesvári (no date) ‘On the role of optimization in double descent: A least squares study’, Advances in engineering education [Preprint].

---

> ### Author Response · Authors · 2022-08-02
> **Answer 3zWt**
>
> We thank you for the thorough review and for appreciating the context and the contributions of our work!
>
> We think your point concerning the NTK is very interesting. We initially wrote the NTK as it seemed straightforward to go from the model $\Phi^\top \Xi (I_n - \gamma P^\pi) \Phi$ to NTK-based that may have looked like $\nabla_0^\top \Xi (I_n - \gamma P^\pi) \nabla_0$ where $\nabla_0$ would have been the gradient at initialization. While we wouldn't have been able to derive a closed form, we think we may still be able to get a fixed-point scalar equation on the Stieljes transform and as such one for the measure which could have been computed numerically. However we are aware there are some well-founded criticisms of the NTK approximation.
>
> However we did not know about the works you mention on the impact of the parameterization mentioned in the two works you mentioned. Thank you very much for the suggestion, we will read them carefully, they could be very good starting point for extending our analysis to neural networks! Once again, the additional dependency on the sampling distribution and the transition kernel of the MDP might complicate the analysis but some numerical solutions may still be within reach.
>
> Thank you for the review and for your suggestions!

---

### Author Response · Authors · 2022-08-02
**General message**

We would like to thank the reviewers for the time and effort they have spent on the paper, we think the points raised are pertinent.
Furthermore we are aware that the paper is technical and at the unusual intersection of RL theory and random matrix/graph theory, so while we tried our best to make it accessible, we welcome suggestions regarding clarity.

While we answer each of the reviewers separately, one point in particular was raised several times so we will address it here.
This point is about the limitations of a model using an asymptotically rank 1 matrix for the transition kernel of the MDP.

A. **Why this choice?**
    While we could for example have done a different set of assumptions, like considering $P^\pi$ symmetric and of rank $k = o(n)$ which would have given us also a complete characterization of the spectrum (it would simply have been a regular spiked MP spectrum as well) it is not clear what this assumption entails for the graph and its dependency on the stationary distribution $d^\pi$. We decided to chose the simplest model that would 1) allows us to have a full characterization of the spectrum and 2) would allow us to model any stationary distribution $d^\pi$.

Regarding 1), we could have made the choice of considering more complex models at the expanse of losing the analytical solution but still able to get an equation satisfied by the spectral measure (see slides 22 of https://random-matrix-learning.github.io/#presentation2) and compute it numerically efficiently. However we would have lost the analytical form and thus all the insights we developed concerning the interplay between the overparameterization ratio $\rho$, the discount factor $\gamma$, the stationary distribution $d^\pi$, the behavior distribution $\xi$ and the reward $r$.
Additionally, note that Prop 3.2 (spectrum of TD) is already non-trivial, as in the literature only the case of symmetric covariance matrices is studied and here $P^\pi$ is non-symmetric. Furthermore the mathematical field of random graph spectral theory is still young (last 2 decades mostly) and we were often limited by the results available.

For 2), it is clear from our results on the peaking behavior of the optimization of TD (Prop 4.2, Fig 2) that the stationary distribution plays an important role in the optimization and thus needs to be taken into account. Therefore this choice of the deformed Erdos-Renyi graph was flexible enough (with $n$ free parameters, $n-1$ for $d^\pi$ and 1 for $\mu$) to totally take $d^\pi$ into account while being simple enough to have an analytical spectrum.

 B. **The model is wrong, is it useful? And why?**
 A counter argument we would like to raise is that while the model is definitely simplistic, for instance the spectrum of $P^\pi$ for a 4 rooms domain in Fig 1.a does not at all satisfy the assumption, we can still expect the results about the extreme eigenvalues to hold (eq 9 and Fig 1.e and 1.f). A property of spiked models (cf Theorem 2.13, https://romaincouillet.hebfree.org/docs/RMT_ML_Book.pdf) is that the order of the spikes is conserved. This means that for two perturbations with magnitude $x_1 < x_2$, their resulting eigenvalues in the spiked MP law satisfy $\lambda_1 <\lambda_2$. This is important as we are only modelling the minimum eigenvalue of $I-\gamma P^\pi$ which is $1-\gamma$ therefore we can expect all other eigenvalues to be above its spike. This is exactly what we see on Fig 1.d. While we do have other eigenvalues beyond the bulk, the smallest one (and thus important one for optimization bounds) is the one associated to the first eigenvalue of $P^\pi$. (Note that the highest eigenvalue will be within a factor at most $1+\gamma$ of the highest eigenvalue in the bulk, and thus doesn't change the optimization rates).
    This illustrates that, while the assumption is not expressive enough to capture the full spectral distribution in a realistic case, it is enough to capture accurately the extreme eigenvalues, which are the important ones for understanding the optimization of TD (Fig 1.e and 1.f).

 C. **Long traces improve connectivity**
    An important point that we did not stress enough in paper is the fact that for any Markov kernel $P^\pi$, we have $\lim_{k\to+\infty} (P^\pi)^k = 1_n d^{\pi \top}$. This means that, as we use longer traces, i.e when doing $k$-step returns as often done in practice, we contract the magnitude of all the eigenvalues $<1$. In particular, if we use $k$-step TD where $k$ grows with $n$, we will satisfy also the asymptotically well-connected property. It seems that in practice (https://arxiv.org/abs/1806.01175) using longer traces is beneficial in richer environments, giving weight to that argument. Indeed, some modern algorithms on rich environments, like IMPALA on DMLab (https://arxiv.org/abs/1802.01561) use long traces, such as $k=100$. This would mean that a second eigenvalue of $0.99$ would be shrunk down to approximately $0.37$.

---

> ### Author Response · Authors · 2022-08-09
> **general message (2)**
>
> We have updated a slightly modified version of the paper where we added the relevant work reviewer EBEx pointed to us. Furthermore we added a few details and we used larger graphs/MDPs in Fig 1 which better shows how accurate our model is, even in MDP where P is high rank.
>
> We hope the rebuttal helped understand our contribution and why some simple assumptions (random features, well-connected graph) could still be very predictive of behaviors seen in more complex MDPs.

---

> ### Author Response · Authors · 2022-08-09
> **On the strength of the random graph assumption**
>
> We would like to come back on a point raised by reviewer EBEx concerning the strength of the assumption: **random feature vs our random graph model and explain why the latter is actually not in itself a stronger assumption, quite the contrary.**
>
> The random feature assumption (very prevalent and well studied) will assume, in its simplest form, each entry of the features are i.i.d. Note that this rules out any modelling of the structure within the features, and the Marchenko-Pastur distribution is simply a consequence of this i.id noise assumption.
>
> In spirit, assuming that the adjacency matrix of our graph is i.i.d is exactly the same kind of assumption as graphs are often represented by their set of edges. However, in our paper, we assume something beyond the i.i.d setting and model the stationary distribution of the graph. The concentration to a eigenvalue at 1 and a disk of magnitude $\propto \frac{1}{\sqrt{n}}$ (for the i.i.d case) is the same type of result as the MP result above.
>
> **Thus, to compare the strength of the two assumptions: if one was given Normal data with a covariance $\Sigma$. Random i.i.d features would simple model it as $\sigma \mathbf{I}$ where $\sigma$ is a scalar while our model for the random graph that takes the stationary distribution into account can be seen as modelling it with a diagonal matrix $diag(\sigma_1, \dots \sigma_n)$.**
>
> Given the wealth of papers and results concerning the superiority of diagonal approximations compared to scalar one, we argue that our modelling assumption for the graph cannot be simply thought unrealistic when compared to the random features assumptions (which has proven accurate and useful theoretically in many domains).
>
> We furthermore believe that this type of analysis (and future extensions of it) will be key to an average-case predictive theory of reinforcement learning in large domains and with large models.

---

### Meta-Review · Area_Chair_HBA9 · 2022-08-26

**Recommendation:** Accept
**Confidence:** Less certain

**Metareview:**

The paper studies policy evaluation with linear TD(0) learning in the over-parameterized regime. Using random matrix and random graph theory, the paper characterizes the spectrum of the TD operator and use this to show that TD learning exhibits a double-descent phenomenon.

The reviewers found this technical paper to be clear and well-written. They also appreciated the novel analysis, extending results on learning in the important over-parameterized regime in supervised learning to reinforcement learning. However, there were also concerns about the limiting assumptions, in particular the asymptotically rank 1 transition matrix. The authors' response provided further discussion and motivation for this choice which the reviewers found convincing. Overall, while this paper does make strong assumptions, it also provides good insights and novel results that are likely to spark further research in this area. As a result, it is recommended to be accepted.

**Award:**

No

---

### Decision · Program_Chairs · 2022-09-14

Accept